# Mediterranean nascent sea spray organic aerosol and relationships with seawater biogeochemistry

Evelyn Freney[1], Karine Sellegri[1], AlessiaNicosia[1] , Matteo. Rinaldi[3], Leah. R. Williams[4], Jonathan T. Trueblood[1], André. S. H. Prévôt[5], Melilotus Thyssen[6], Gérald Grégori[6], Nils Haëntjens[7], Julie Dinasquet[8, 9*], Ingrid Obernosterer[9], France Van Wambeke[6], Anja Engel[10], Birthe Zäncker[10], Karine Desboeufs[11], Eija Asmi[2], Hilkka Timonen[2], Cécile Guieu[12]

[1] Université Clermont Auvergne, CNRS, Laboratoire de Météorologie Physique (LaMP) F-63000 Clermont-Ferrand, France

[2] Atmospheric Composition Research, Finnish Meteorological Institute, Helsinki, FI-00101, Finland

[3] National Research Council (CNR), Institute of Atmospheric Sciences and Climate (ISAC), Bologna, Italy

[4] Aerodyne Research, Inc., Billerica, Massachusetts, USA

[5] Laboratory of Atmospheric Chemistry, Paul Scherrer Institute, 5232 Villigen PSI, Switzerland

[6] Aix-Marseille Université, Université de Toulon, CNRS, IRD, Mediterranean Institute of Oceanography UM110, Marseille 13288, France

[7] School of Marine Sciences, University of Maine, Orono, ME 04469, USA

[8] Marine Biology Research Division, Scripps Institution of Oceanography, 92037 La Jolla, US

[9] CNRS, Sorbonne Université, Laboratoire d'Océanographie Microbienne, UMR7621, F-66650 Banyuls-sur-Mer, France

[10] GEOMAR, Helmholtz Centre for Ocean Research Kiel, 24105 Kiel, Germany

[11] LISA, CNRS UMR7583, Université Paris Est Créteil (UPEC), Université de Paris, Institut Pierre Simon Laplace (IPSL), Créteil, France

[12] Sorbonne Université, CNRS, Laboratoire d'Océanographie de Villefranche, LOV, F-06230 Villefranche-sur-Mer, France

*now at Center for Aerosol Impacts on Chemistry of the Environment, CASPO, Scripps Institution of Oceanography, UCSD, USA

*Correspondence to*:evelyn.freney@uca.fr

**Abstract**

The organic mass fraction from sea spray aerosol (SSA) is currently a subject of intense research. The majority of this research is dedicated to measurements in ambient air, although recently a small number of studies have additionally focused on nascent sea spray aerosol. This work presents measurements collected during a five-week cruise in May and June, 2017 in the central and western Mediterranean Sea, an oligotrophic marine region with low phytoplankton biomass. Surface seawater was continuously pumped into a bubble bursting apparatus to generate nascent sea spray aerosol. Size distributions were measured with a differential mobility particle sizer (DMPS). Chemical characterization of the submicron aerosol was performed with a time of flight aerosol chemical speciation monitor (ToF-ACSM) operating with a 10-minute time resolution, and with filter-based chemical analysis on a daily basis. Using positive matrix factorization analysis, the ToF-ACSM non-refractory organic matter ($OM_{NR}$) was separated into four different organic aerosols types, identified as primary OA

(POA$_{NR}$), oxidized OA (OOA$_{NR}$), a methanesulfonic acid type OA (MSA-OA$_{NR}$) and a mixed OA (MOA$_{NR}$). In parallel, surface seawater biogeochemical properties were monitored providing information on phytoplankton cell abundance and seawater particulate organic carbon (one-hour time resolution), and seawater surface microlayer (SML) dissolved organic carbon (DOC) (on a daily basis). Statistically robust correlations (for n > 500) were found between MOA$_{NR}$ and nano-phytoplankton cell abundance, as well as between POA$_{NR}$, OOA$_{NR}$ and particulate organic carbon (POC). Filter-based analysis of the submicron SSA showed that the non-refractory organic mass represented on average only 40% of the total organic mass, which represents approximately 22% of the total sea spray mass. Parameterizations of the contributions of different types of organics to the submicron nascent sea spray aerosol are proposed as a function of the seawater biogeochemical properties for use in models.

## 1 Introduction

Oceans cover approximately 70% of the Earth's surface and sea spray emissions contribute up to 6 kTons/Yr of particulate matter, making them a major primary source in the atmosphere (Lewis and Schwartz, 2004). The majority of the mass associated with sea spray emissions is in the form of coarse mode sea salt particles. However, it is now well known that the submicron fraction of marine emissions is also important and contains a significant portion of organic compounds (Facchini et al., 2008). This organic fraction tends to be highest during phytoplankton bloom events (O'Dowd et al., 2004). Although the organic fraction of the aerosol population represents little mass, the high number concentration of these aerosol particles makes them a significant contributor to the potential cloud condensation nuclei concentration (Burrow et al., 2018). Organics in sea spray have also been shown to contribute to potential marine ice nuclei (McCluskey et al., 2017). Understanding how the organic fraction of marine aerosol particles transfer to the atmosphere is essential to help identify the contribution of the marine aerosols to the Earth's radiative budget.

In general, sea spray aerosol is generated through bubble bursting after wave breaking at the ocean surface, a process that has been described in early publications (Blanchard and Woodcock, 1980). Once bubbles burst, film and jet droplets are ejected into the atmosphere, and dry to leave aerosol particles as residues containing the different constituents of the bulk seawater and surface microlayer. The size distribution of this sea spray extends from the fine particle range, with diameters < 100 nm to the super-micron range (up to 3 μm). The super-micron range is known to be mainly composed of refractory NaCl and different MgCO$_3$ or (Mg)$_2$SO$_4$ species, while the submicron fraction tends to be enriched with organic compounds (O'Dowd et al., 2004).

Traditionally, measurements in and around the marine boundary layer were made using offline filter measurements followed by laboratory-based analysis using either ion chromatography or organic carbon and elemental carbon analysis. However, over the last decade, there has been a significant increase in the number of studies deploying online aerosol mass spectrometry methods, including laser ablation mass spectrometry (Dall'Osto et al., 2019) and thermal vaporization followed by electron impact ionization methods (Giordano et al., 2017; Ovadnevaite et al., 2011; Schmale et al., 2013). The aerosol mass spectrometer (AMS, Aerodyne Research, Inc) and aerosol chemical speciation monitor (ACSM) are examples of the latter type and are widely used instruments to monitor the chemical composition of submicron particulate matter in the atmosphere. The design of this instrument is optimized to investigate the non-refractory fraction, defined as material that

vaporizes at 600°C of aerosol mass in the atmosphere. In the majority of atmospheric environments, the submicron fraction of the aerosol is dominated by non-refractory organic and inorganic species. In marine environments, the aerosol mass spectrometry has been used to better characterize the chemical properties of ambient marine aerosol particles, from the Atlantic (Ovadnevaite et al., 2011), to Antarctic coastal environments (Schmale et al., 2013; Giordano et al., 2017). Using a combination of high-resolution aerosol mass spectrometry and positive matrix factorization analysis, different marine organic aerosols have been identified including secondary marine organic aerosols, methanesulfonic acid (MSA), containing organic aerosols, and amino acid (AA) associated organic aerosols. The latter were thought to be primarily linked to local sea life emissions at the measurement site (Schmale et al. 2013).

Most of these studies measured ambient aerosol already modified through atmospheric chemical and physical processes. Current knowledge on the source and evolution of nascent sea spray organic emissions is still limited. This is attributed to the natural variability of marine organic aerosol and to the lack of high temporal resolution studies at the ocean/atmospheric interface. A limited number of studies have focused directly on the composition of nascent sea spray aerosol particles emitted from wave action in controlled simulation chambers (Wang et al., 2015, Collins et al., 2016), or through dedicated bubble bursting experiments in ambient environments (Bates et al., 2012; 2020; Dall'Osto et al., 2019; Park et al., 2019). These studies in controlled environments identified the presence of aliphatic-rich and amino acid-rich organic aerosols related to different phases of phytoplankton blooms (Bates et al. 2012; Wang et al., 2015). Dall'Osto et al. (2019) identified an amino acid contribution in both nascent sea spray aerosol and ambient aerosols. Park et al, (2019) observed that sea-salt aerosol production was positively correlated with organic compounds in the water, notably dissolved organic carbon, marine microgels and chloropyll a (Chl-a). However, in all of these studies, samples were collected at point intervals and were not able to provide information on the evolution of aerosol physical and chemical properties over a large spatial area. In addition to this, previous studies do not provide measurements making it possible to provide the quantitative link between seawater biogeochemistry and the SSA organic composition.

In this work, we characterized the chemical composition of the nascent sea spray submicron aerosol continuously generated from the underway seawater system of the *R/V Pourquoi Pas?* over a five-week campaign in the Mediterranean Sea. The Mediterranean Sea is a low nutrient low chlorophyll (LNLC) environment and was characterized by oligotrophic conditions along the whole field campaign (Guieu et al., 2020). Understanding the formation of nascent sea spray aerosols in such an LNLC system can provide valuable information and be extrapolated to other oligotrophic environments.

**2-Methodology**

**2.1 The PEACETIME oceanographic campaign**

The French research vessel, the *R/V 'Pourquoi Pas?'*, was deployed for a five-week-long period from the 10[th] of May to the 10[th] of June 2017 on the Mediterranean Sea, as part of the project: PEACETIME (ProcEss studies at the Air-sEa Interface after dust deposition in The Mediterranean sea) project. The ship track (Fig. 1) started and ended in La Seyne-sur-Mer, France. The ship traveled clockwise covering latitudes in the Mediterranean Sea from 35° to 42°, and longitudes from 0° to 21°.

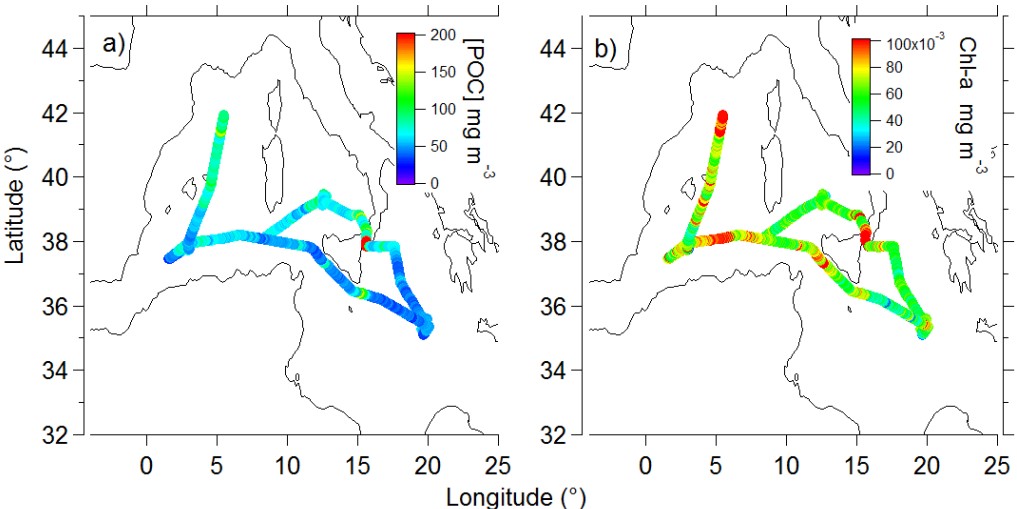

**Figure 1: The ship track in the Mediterranean Sea during the PEACETIME expedition. The trajectory is colored by a) POC b) *Chl-a*.**

Results from a suite of hydrology and biogeochemistry measurements performed on board are given in Guieu et al. (2020). In addition to standard seawater temperature (T) and salinity (S) measurements, the concentrations of a wide range of chemical and microbiological parameters were monitored hourly. Several plankton functional groups were identified, including Synechococcus, Prochlorococcus, nanoeurkaryotes, Coccolithophores-like, Cryptophytes-like, and microphytoplankton. The sea surface temperature (T) showed a gradual increase from the start to the end of the campaign from 19°C up to 23°C. Sea surface salinity (S) varied from 36 up to 39 PSU increasing from east to west. The sampling region was characteristic of open sea (average depth 2750 m ± 770 m along the transect). The sea was calm to moderately rough throughout the sampling period with conditions always remaining below Beaufort 4. Wind speed varied between 10 and 20 m s$^{-1}$.

Total chlorophyll-a (*Chl-a*) and particulate organic carbon (POC) were also measured. In line with the oligotrophic state of the Mediterranean Sea during this period, the POC concentrations were highest at the most northern lontitudes and gradually decreased along the ship transect (Fig. 1a), while the *Chl-a* concentration remained stable and low (0.07 ± 0.03 mg m$^{-3}$) throughout the sampling period (Fig. 1b).

### 2.2. Surface Seawater Analysis

### 2.2.1. Flow cytometry

Phytoplankton cells were counted with one-hour time resolution using an automated Cytosense flow cytometer (Cytobuoy, NL) connected to a continuous clean pumping underway seawater system, as described in Thyssen et al. (2010) and Leroux et al. (2017). Particles were brought within a laminar flow filtered seawater sheath fluid, and detected with forward scatter (FWS) and sideward scatter (SWS) as well as fluorescence in the red (FLR >652 nm) and orange (FLO 552-652 nm) in the size range 1- 800 μm. Two trigger levels were applied for the distinction between highly concentrated picophytoplankton and cyanobacteria groups (trigger level FLR 7.34 mV, sampling at a speed of 4 mm$^3$s$^{-1}$ and analyzing 0.65 ± 0.18 cm$^3$), and lower concentrated nano- and micro phytoplankton (trigger level FLR 14.87 mV, at a speed of 8 mm$^3$s$^{-1}$ and analyzing 3.57 ± 0.97 cm$^3$). Different sets of 2D projections were plotted in Cytoclus® software to manually gate phytoplankton groups. To follow

stability of the flow cytometer, 2 μm red fluorescing polystyrene beads (Polyscience) were regularly analyzed. The use of silica beads (1, 2, 3, 5, 7 μm in diameter, Bangs Laboratory) for size retrieving estimates from FWS were used to separate picoplankton from nanoplankton clusters.

### 2.2.2. Chlorophyll-a and POC

From the underway seawater system, *Chl-a* was derived from the particulate absorption spectrum line-height at 676 nm (Bloss et al., 2013) after the relationship was adjusted to PEACETIME *Chl-a* derived from HPLC (*Chl-a* $= 194.41 \times$ line_height$^{1.131}$). POC was estimated from particulate attenuation at 660 nm using an empirical relationship specific to PEACETIME (POC $= 1405.1 \times c_p(660) - 52.4$) which was slightly higher than the literature value likely due to the small dynamic range (1.27 higher on average for the range observed (Cetinic et al., 2012)). Particulate attenuation and absorption of surface water were measured continuously with a WetLabs Spectral Absorption and Attenuation Meter using a flow-through system similar to the setup described in Slade et al. (2010). Both the *Chl-a* and the POC were obtained with a time resolution of 1 minute.

### 2.2.3. Heterotrophic bacteria counts and bacterial production

For the enumeration of heterotrophic bacteria, discrete samples collected using the Niskin surface bottle (< 5 m), subsamples (4.5 mL) were fixed with glutaraldehyde grade I 25% (1% final concentration), and incubated for 30 min at 4 °C, then quick-frozen in liquid nitrogen and stored at -80 °C until analysis. Samples were thawed at room temperature. Counts were performed on a FACSCanto II flow cytometer (Becton Dickinson) equipped with 3 air-cooled lasers: red (633 nm), blue (488 nm), and violet (407 nm). For the enumeration of heterotrophic bacteria, cells were stained with SYBR Green I (Invitrogen – Molecular Probes) at 0.025% (vol / vol) final concentration for 15 min at room temperature in the dark. Stained cells were discriminated and enumerated according to their right-angle light scatter (SSC) and green fluorescence using a 530/30 nm bandpass filter. In a plot of green versus red fluorescence, heterotrophic bacteria were distinguished from autotrophic prokaryotes. Fluorescent beads (1.002 μm; Polysciences Europe) were systematically added to each analyzed sample as an internal standard. The cell abundance was determined from the flow rate, which was calculated with TruCount beads (BD Biosciences).

Heterotrophic prokaryotic production (BP) was estimated from rates of [3]H leucine incorporation using the micro centrifugation technique. The detailed protocol is available in Van Wambeke et al. (2020). Briefly, triplicate 1.5 mL subsamples from the Niskin surface bottle (< 5 m) and one blank were incubated in the dark at in situ temperature. Leucine was added at 20 nM final concentration and the leucine - carbon conversion factor used was 1.5 kgC mol$^{-1}$. These samples were collected at 25 specific stations along the ship transect.

### 2.3. Surface MicroLayer (SML) Sampling and Analysis

### 2.3.1. Sampling

Surface microlayer SML sampling was conducted twice a day from a zodiac using a 50 x 26 cm silicate glass plate sampler (Harvey 1966; Cunliffe and Wurl 2014) with an effective sampling surface area of 2600 cm$^2$ considering both sides. For sampling, the zodiac was positioned 0.5 nautical miles away from the research vessel and into the wind direction to avoid contamination. The glass plate was immersed perpendicular to the sea surface and withdrawn at ~17 cm s$^{-1}$. SML samples were removed from the plate using a Teflon wiper (Cunliffeand Wurl, 2014) and collected in an acid cleaned and rinsed bottle. Prior to sampling, all equipment

was cleaned with acid (10 % HCl), rinsed in MilliQ water and copiously rinsed with seawater directly before samples were taken.

### 2.3.2. DOC analysis

The concentration of dissolved organic carbon (DOC) was determined in samples filtered online (Sartoban © 300; 0.2 μm filters). Subsamples of 10 mL (in duplicate) were transferred to pre-combusted glass ampoules and acidified with $H_3PO_4$ (final pH = 2). The sealed glass ampoules were stored in the dark at room temperature until analysis. DOC measurements were performed on a Shimadzu TOC-V-CSH (Benner and Strom, 1993). Prior to injection, DOC samples were exposed to $CO_2$ -free air for 6 min to remove inorganic carbon. A 100 μL aliquot of samples was injected in triplicate and the analytical precision was 2%. Standards were prepared with acetanilide. Analysis of DOC was performed both on SML and the underlying seawater sampled from the zodiac.

## 2.4. Seaspray generation and analysis

### 2.4.1 General set-up

The sea spray generator has been characterized and deployed in a number of previous studies and full details are reported in Schwier et al. (2015). Briefly, it consists of a 10 L glass tank, fitted with a plunging jet system for the water. A particle-free air flushing system, placed perpendicular to the water surface at a distance of 1 cm to send a constant airflow across the surface of the water to replicate the effects of wind on the surface of seawater (13 m $s^{-1}$). The sea spray generator was supplied with a continuous flow of seawater collected at a depth of 5 m by an underway seawater circulating system operated with a large peristaltic pump (Verder® VF40 with EPDM hose). The waste water was evacuated downstream of the sampling location to avoid any contamination.

The various aerosol instrumentation, including a time of flight aerosol chemical speciation monitor (ToF-ACSM, (Aerodyne)), a custom made differential mobility particle sizer (DMPS) coupled with a condensation particle counter (CPC, model 3010 (TSI)), and an impactor (Dekati, PM1) collected submicron particulate matter for offline ion chromatography and chemical analysis, sampled from the headspace above the seawater in the tank. A silica gel dryer was connected to the output of the chamber, which was subsequently connected to a flow dispatcher which had two outputs of equal length (< 50 cm), one to the ToF-ACSM and the second to a DMPS-CPC. The aerosol relative humidity was measured continuously and varied from 20% to 40% (Fig. S1). The total sampling line length was approximately 2 m with a sampling flow of 5 L/min giving a residence time of less than 30 seconds. A schematic of the sampling setup is shown in Fig. S2. Regular tests were performed to ensure that the system was airtight and free from external aerosol influences.

### 2.4.2 Aerosol physical and chemical properties

#### Size distribution measurements

Particle size distribution and number concentration measurements were obtained using the DMPS-CPC. Measurements were provided approximately every 10 minutes for 25 different size classes ranging from 10 nm up to 500 nm. The size distribution was relatively constant throughout the measurement period giving a principal size mode at 110 nm and a second mode at 300 nm. This size distribution is characteristic of the bubbler

seawater generation method (Schwier et al., 2015), and is similar to that from other nascent seawater aerosol generators (Bates et al., 2012), and to that observed in the clean marine boundary layer (Yoon et al., 2007). Although the size distribution remained constant, the absolute number concentration varied by a factor of 3 over the sampling period. Details of these changes in aerosol number concentration as well as the associated cloud condensation nuclei activity are detailed in a companion paper (Sellegri et al., 2020).

**Offline PM1 filter analysis**

In parallel to the online aerosol physical and chemical measurements, the generated nascent sea spray aerosol particles were also sampled onto PM1 quartz filters. Aerosol samples were extracted in MilliQ water by sonication (30 min) for the analysis of the water-soluble components. Extracts were analyzed by ion chromatography for the quantification of the main inorganic ions (Sandrini et al., 2016). An IonPac CS16 3 × 250 mm Dionex separation column with gradient MSA elution and an IonPac AS11 2 × 250 mm Dionex separation column with gradient KOH elution were deployed for cations and anions, respectively. The contributions of the estimated $ssSO_4$ (seasalt $SO_4$), ssK, ssMg, ssCa, and ssCl were calculated based on the seawater theoretical ratio (Seinfeld and Pandis, 2006 (These ratios are: $SO_4$: 0.25, K: 0.06, Mg: 0.12, Ca:0.04)). The remaining nss fraction of the inorganic aerosol was within the measurement error of the instrument.

The water-soluble organic carbon (WSOC) content of the extracts was quantified using a total organic carbon (TOC) thermal combustion analyzer (Shimadzu TOC-5000A) (Detection limit (DL) = 1.9 μgC/filter). Measurements of the total carbon (TC) content were performed on a filter punch cut before water extraction by a thermal combustion analyzer equipped with a furnace for solid samples (Analytik Jena, Multi NC2100S; Rinaldi et al., 2007) (DL = 37.9 μgC/filter). For the organic carbon (OC) analysis, the punch was acidified before analysis to remove inorganic carbon from TC and obtain OC. This process included positioning the punch in the instrument sample container, covered with 40 μL of $H_3PO_4$ (20% w/w) and left under oxygen flow, at room temperature, for ca. 5 minutes to allow the volatilization of carbonates as carbon dioxide ($CO_2$). The process was monitored online by the analyzer NDIR $CO_2$ detector: When the $CO_2$ level went back to baseline conditions, the vessel was placed into the furnace (950°C) for the OC analysis.

The water insoluble organic carbon (WIOC) is not measured directly but is derived from the difference between WSOC and OC. As mentioned, the measurements of OC and WSOC are made with two different instruments with that for OC having a much lower limit of detection (DL). Although, the quantification of WSOC was always possible, some samples had OC concentrations < DL. Assuming that OC=WSOC (and WIOC = 0) would be incorrect andwould result in significant error in the estimate of total OC. There could be an important amount of WIOC that we cannot quantify because of the lower sensitivity of the OC analysis. For this reason, we presented WIOC and OC data only for samples that have both quantifiable OC and WSOC.

The amount of inorganic carbon varies between 12 and 71% of TC, and thus the acidification process for sea spray is an important step to follow for TOC measurements. Organic carbon was converted to organic mass using conversion factors of 1.4 for the conversion of WIOC to WIOM, and 1.8 for the conversion of WSOC and WSOM, respectively (Facchini et al., 2008).Although filters were only collected on a daily basis, they provided

valuable information on the refractory component of the aerosol population. The volume concentrations measured on the filters were similar to those calculated from the DMPS (Fig. S3).

**ToF-ACSM**

The ToF-ACSM is based on the same operating principles as the aerosol mass spectrometer (Drewnick et al., 2009). TheToF-ACSM contains a critical orifice, a PM1 aerodynamic lens to focus submicron particles into a narrow beam that flows into a differentially pumped vacuum chamber, a standard vaporizer heated (600$^\circ$C) to vaporize particles, an electron emitting tungsten filament (70eV) to ionize the vapor, a compact time-of-flight, mass analyzer (ETOF, TOFWERK AG, Thun, Switzerland) and a discrete dynode detector (Fröhlich et al., 2013). It does not have the ability to size aerosol particles but has the advantage of being more compact and more robust for continuous observations than the AMS (Fröhlich et al., 2013). The ToF-ACSM alternates between sampling ambient air and sampling through a filter in order to subtract the signal due to air.

During this experiment the ToF-ACSM was operated in a 2-min filter and 8-min sample mode with a measurement every 10 minutes. The aerodynamic lens transmits particles between 70 nm and 700 nm, making the ACSM approximately a PM1 measurement. The non-refractory particle material (NR-PM) is defined similarly as in DeCarlo et al. (2006) as aerosol particles that are vaporized using the 600$\circ$C resistively heated vaporizer and detected during the instrument sampling interval. The relative ionization efficiency for $NH_4$ was 3.12 and for $SO_4$ 0.8, determined from calibrations with ammonium nitrate and sulfate. However, considering that the measured $SO_4$ concentrations represent seasalt(ss) $SO_4$, we adjusted the RIE of $SO_4$ to 0.3. Using this value we observe that the mass concentration of $SO_4^{2-}$ from the ACSM follows that of the filters for the start of the field campaign, but is lower than the filter $SO_4$ concentration near the end of the campaign. The temperature of the vaporizer and the size range do not permit efficient detection of sea salt particles. However, in situations of high sea salt concentrations, detection of sea salt ions and related halides have been reported (Bates et al., 2012; Giordano et al., 2017; Ovadnevaite et al., 2011; Schmale et al., 2013, Timonen et al., 2016). Likewise, in this study mass spectral signals associated with sea salt were observed. In addition, the contribution from chloride was very high (72% of the total mass). In some quadrupole ACSM instruments, negative Chl peaks are often observed (Tobler et al., 2020), due to slow evaporation of refractory material from the vaporizer relative to the 30 s switching time of filter and sample. This tends to overestimate the filter measurement and underestimate the sample measurement and can lead to negative values for the difference. However, during these measurements with the ToF-ACSM, negative Chl was not observed due to the long switching times.

The typical signature peaks for sea salt aerosol in our instrument were confirmed by atomizing pure aerosol particles generated from sea salt solution (Biokar, synthetic sea salt, lot: 0017475), passing the particles through a silica gel dryer and into the ToF-ACSM instrument. In the default fragmentation table used to assign the signals at individual m/z's to chemical species (Allan et al., 2004), peaks associated with sea salt were identified as organic aerosol fragments. In order to better represent the measured aerosol composition, we modified the standard fragmentation table by introducing a sea salt species that includes m/z fragments at m/z 23 ($Na^+$), m/z 35 and 37 ($^{35}Cl^-$, $^{37}Cl^-$), 58 and 60 ($Na^{35}Cl$, $Na^{37}Cl$), and 81 and 83 ($NaCl_2$, $NaCl^{37}Cl$). For m/z 81, there is overlap with an $SO_4$ fragment and a correction suggested by Schmale et al. (2013) was applied (Eq.1). This correction accounted for less than 10% of the signal at m/z 81 and 3% of the total sulfate signal.

frag_SO4[81] = 81−frag_organic[81] −0.036×frag_Na[23]                                        (Eq.1)

Quantification of sea salt is difficult in the ToF-ACSM due to inefficient vaporization and a nonlinear contribution to the Na$^+$ signal from surface ionization on the vaporizer. Therefore, in this work we do not attempt to quantify the sea salt fraction, but instead use the mass spectral information to separate it from the organic aerosols. A standard collection efficiency (CE) of 0.5 was applied to all data obtained from the ACSM (Middlebrook et al., 2012). Regular particle free sampling periods were performed to ensure that there was no buildup of material on the vaporizer and that the sampling set up was leak-free.

**Positive matrix factorization (PMF)**

In order to identify the different organic aerosols present in the sea spray from primary seawater, unconstrained positive matrix factorization, using the SoFi interface (Canonaco et al., 2013) was performed on the ToF-ACSM organic mass spectra. The ToF-ACSM gives unit mass resolution (UMR) mass spectra, so it is not possible to distinguish between salt and non-salt ions at a given m/z. A decision was made to remove all sea salt related ions from the organic mass spectral data matrix, giving a total of 116 m/z from 0 up to 150. We are aware that removing the m/z's associated with NaCl (23, 35, 37, 58, 60, 81 and 83) will also remove contributions from organics at these m/z's. However, the organic contributions at these m/z values are small relatively to the rest of the organic MS and are typically a factor of 10 smaller than sea salt in the ACSM signal.

During sampling, the ToF-ACSM was run with a 2 min filter / 8 minute cycle. When sampling with long times between filters, any drift in sensitivity can result in a difference signal that is an artefact. This is especially true for those signals with a high background (e.g., from air at m/z 29 from $^{15}$NN), and when measuring low concentrations of organic mass (as is the case during this experiment). In PMF solutions including m/z 29, one of the factors contained predominantly m/z 29 and the time series was noisy and flat. Downweighting m/z 29 (by x100) did not help distribute the signal at m/z 29 into the other organic factors. Therefore, since m/z 29 was only contributing to noise and not to chemical information, we chose to remove it. The PMF solutions were explored up to six factors, as a function of fpeak values from -1 to +1 (Fig. S4, S5). The four-factor solution was chosen, based on correlations with reference mass spectra, and correlations with external time series. The correlations for three to five factor solutions are illustrated in the supplementary material l (Fig. S6 to S10). The four identified factors, as well as their mass spectral fingerprints and time series will be discussed in the following sections.

**3. Results and discussions.**

**3.1. Time evolution of the chemical composition of nascent sea spray**

Mass concentrations of aerosol chemical composition obtained from the submicron offline filter measurements are listed in Table 1. The soluble inorganic species concentrations were mostly found with proportions similar to the reference average seawater composition (Seinfeld and Pandis, 2006). However, enrichment in K$^+$ (69% of which was not explained by the average seawater composition) and a slight enrichment of Ca$^{2+}$ was measured toward the end of the campaign (17% of Ca$^{2+}$ was not explained by reference seawater composition from the 28$^{th}$ of May onward). In contrast, the magnesium was slightly depleted (20% less than expected in reference seawater composition) but less towards the end of the campaign. Filter-based organic matter (OM) was evenly split between WIOM (14% ± 5% of total mass) and WSOM (9% ± 5% of total mass), which contrasts with previous studies where organic matter in ambient marine aerosol was almost exclusively composed of WIOM (Facchini et

al., 2008). However, these previous studies were conducted during phytoplankton bloom events of the Northern Atlantic Ocean, where POC is usually enhanced. Considering that the Mediterranean Sea is characterized by oligotrophic conditions during PEACETIME, it could explain the relatively low contributions of WIOM.

**Table 1. Concentrations of different chemical species in PM1 primary seawater aerosols measured using offline analysis of filters and online measurements from the ACSM.The cited uncertainty represents 1σ.**

| Offline analysis of filters ($\mu g\ m^{-3}$) | | % | Non-refractory PM1 (ACSM) ($\mu g\ m^{-3}$) | | % |
|---|---|---|---|---|---|
| $ssSO_4^{2-}$ | $1.22 \pm 0.48$ | 5.9% | $SO_4$ | $0.78 \pm 0.34$ | 8.6% |
| $Na^+$ | $4.86 \pm 1.9$ | 23.4% | $NO_3$ | $0.02 \pm 0.02$ | 0.2% |
| $ssCl-$ | $8.75 \pm 3.47$ | 42.1% | $NH_4$ | $0.04 \pm 0.11$ | 0.5% |
| $ssCa^{2+}-$ | $0.18 \pm 0.07$ | 0.87% | Seasalt | $6.7 \pm 5$ | 82% |
| $ssK^+$ | $0.03 \pm 0.01$ | 0.14% | Org | $0.67 \pm 0.26$ | 8.2% |
| $Mg^+$ | $0.59 \pm 0.23$ | 2.8% | | | |
| WIOM | $3.13 \pm 1.12$ | 15.1% | | | |
| WSOM | $2.02 \pm 1.26$ | 9.7% | | | |

The chemical composition of SSA measured by the ACSM is shown in Fig. 2 and listed in Table 1 and was primarily composed of Sea Salt aerosol (determined from the signals at m/z 23 ($Na^+$), 35 ($Cl^-$), 37($Cl^-$) 58($NaCl^+$), 60($NaCl^+$), 81($NaCl)Na^+$ representing 84% ± 15 %) followed by $SO_4$concentrations at 8.6 %, and 8.2% organic matter. The variability in the different chemical compositions throughout the sampling is thought to be a result of the differing associated contributions of refractory compounds ($Ca^{2+}$, $Mg^+$, $K^+$etc) in the seasalt sample.

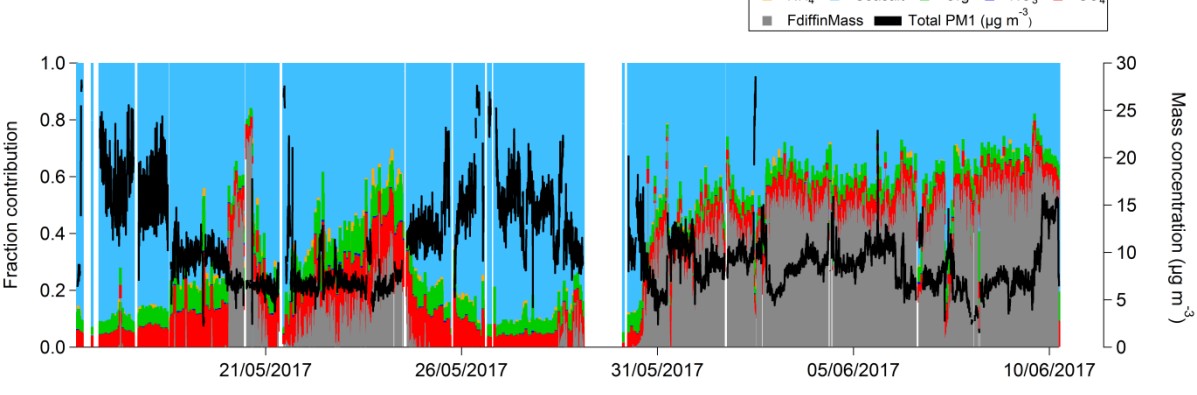

**Figure 2: Time series of the fractional contribution of different species to the ToF-ACSM signal, as well as the total mass concentrations measured by the ACSM (black), and the missing fraction (Refr) in grey.**

In order to determine how representative the ACSM PM1 measurements were of the total PM1 mass, the total ACSM PM1 mass concentration was converted into volume concentration (dividing organic mass concentrations by a density value of 1.2 g cm$^{-3}$, and the other remaining components SO$_4$, NH$_4$, Chl, and NaCl by 2.165 g cm$^{-3}$ (Seinfeld and Pandis, 2006)).This value was compared to the volume concentration measured by the DMPS-CPC, giving a correlation (r) of 0.55, and slope (b) of 1.34 (Fig. 3). The variation in the total seaspray mass concentration with time is a result of the variability in the number of seaspray particles generated from the sea spray generation device. The size distribution of the aerosol remained stable throughout the experiment, but the sea spray number emission flux is influenced by the variability in seawater biogeochemical properties. This is presented and discussed in detail in a companion paper (Sellegri et al., 2020), where we show a clear relationship between the sea spray number concentrations generated in the bubble bursting system and the nanophytoplankton concentration of the seawater, not only in the PEACTIME experiment but also in other seawater types.

The agreement between the two instruments is relatively good in the first part of the campaign, but the difference between the ACSM-derived volume and the DMPS-derived volume increases during the later parts of the campaign from the 1$^{st}$ to the 10$^{th}$ of June (Fig. 3). The fraction of missing mass, Refr, is estimated as the difference between the DMPS volume and ACSM volume divided by the DMPS volume and is shown as a grey area in Fig. 2. The latter period of the field campaign corresponds to a change in seawater type with more inputs from the Atlantic Ocean occurs, and where a corresponding decrease in practical seawater salinity is observed (Fig. S11). It is possible, that during this sampling period the seawater contains higher fractions of refractory material (Refr.) that are less efficiently measured by the ACSM. A second way to estimate the mass missed by the ACSM is the substract the total ACSM mass loading from the total mass measured on the filters. Comparing this estimation of missing mass to the different species measured on the filters shows the best correlations with Mg$^{2+}$, Ca$^{2+}$, and SO$_4^{2-}$ (Fig. S12). This might suggest that the ability of the ACSM to measure NaCl particles depends on how NaCl is associated with other compounds in the sea spray.

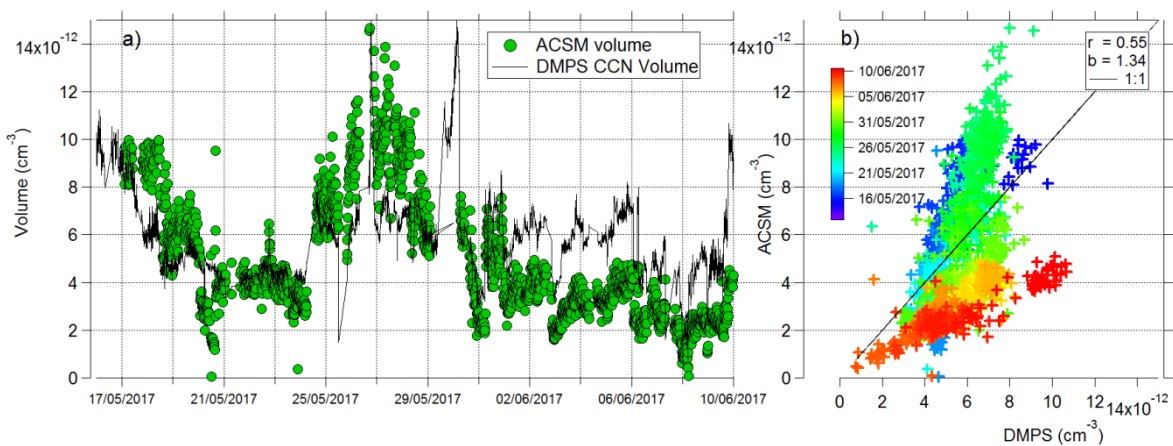

**Figure 3 Comparison between the ACSM and the DMPS volume concentration (#/cm cm$^{-3}$).**

The PM1 mass concentrations of OM calculated from the filters were additionally compared with the total OM measured from the ACSM (OM$_{ACSM}$) (Fig. S13). The OM$_{ACSM}$ represented on average 27% of the total filter OM. 40% WIOM, and 68% of WSOM. This suggests that the differences observed between the filter and

OM<sub>ACSM</sub> measurements are mostly insoluble OM particles internally mixed with the unsampled "Refr" particle, and hence not measured by the ACSM.

OM$_{ACSM}$ varied from 0.2 to 1 μg m$^{-3}$, with higher concentrations measured in the North Western part of the ship track. In the following section, we analyze in more detail the different organic aerosol species present in the SSA samples and determine to what extent they are related to seawater biogeochemical properties.

### 3.2. Marine organic aerosol speciation

As explained in section 2 (Methodology) PMF was used to separate organic factors. Based on the correlations with reference mass spectra, and on observations of the temporal variations we chose a four factor solution. These factors include an oxidized organic aerosol (OOA$_{NR}$), similar to ambient reference OOA (Ng et al., 2010), somewhat oxidized OA containing mixed amino acid and fatty acid signatures, (mixed OA, MOA$_{NR}$), primary organics containing aliphatic signature peaks as well as several peaks corresponding to fatty acids signatures (POA$_{NR}$), and a methanesulphonic acid-like OA (MSA-OA$_{NR}$) (Phinney et al., 2006)(Fig. 4). The correlations of each of the identified factors with reference mass spectra are illustrated in Fig. S8. The OOA$_{NR}$ contributed 51% ± 2% to OA and had signature peaks with high m/z 44 and m/z 28, and correlated with an OOA reference mass spectrum (r = 0.98). It did not contain any other m/z values that might suggest a contribution from other species. The O/C ratio of the OOA$_{NR}$ fraction was 1.6 (calculated using the method described in Canagaratna et al., 2015), which is significantly higher than the average O/C ratio of LV-OOA found in terrestrial ambient aerosols (0.8) (Canagaratna et al., 2015). High O/C ratios have been reported in ambient studies where carbonate species were thought to be measured by the ACSM (Bozzetti et al., 2017, Vlachou et al., 2019), and additionally have been associated with aerosols subjected to aqueous phase processing (Canagaratna et al., 2015).It is also possible that the PMF analysis wrongly attributed excess m/z44 to OOA at the expense of other species such as the MSA-OA$_{NR}$ discussed below. This would impact the reported absolute concentrations of the OOA$_{NR}$ vs the MSA-OA$_{NR}$.

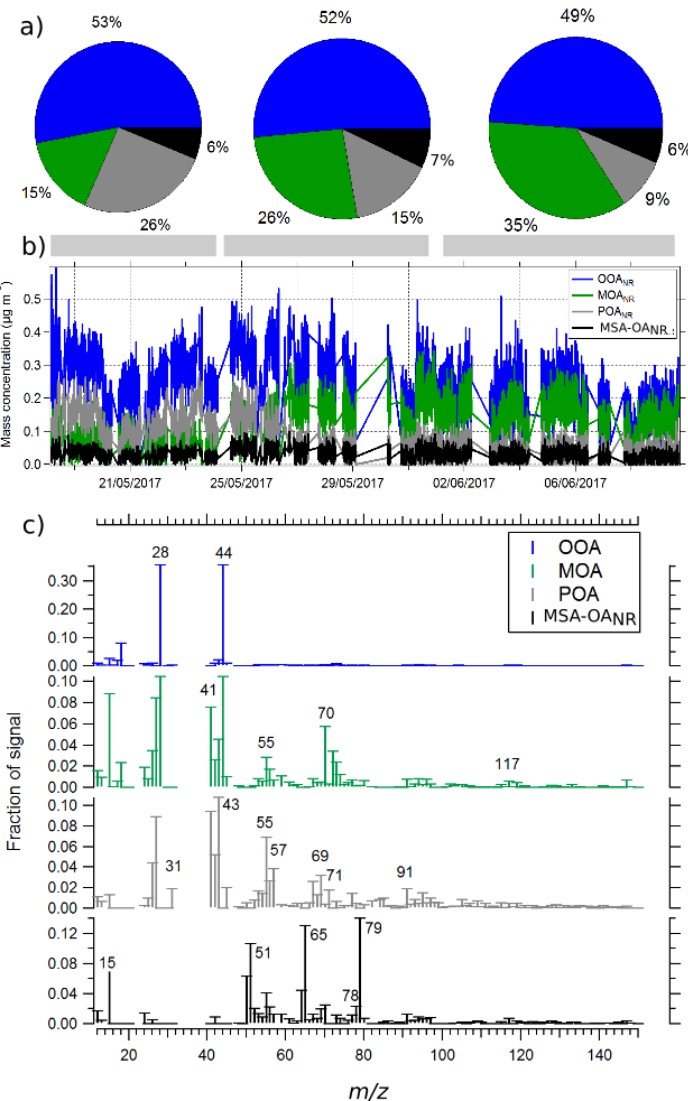

**Figure 4 a) The contribution of the different organic factors during different periods of the PEACETIME ship campaign (identified by the grey bars on the top of b), b) The mass concentrations of each factor (OOA$_{NR}$, MOA$_{NR}$, POA$_{NR}$ and MSA OA$_{NR}$) as a function of time, c) the mass spectra of the factors.**

The second most dominant species was defined as a marine organic aerosol (MOA$_{NR}$). This factor contributed to 15% of the total OA$_{NR}$ at the start of the campaign and then increased to 28% and 35% later on in the campaign (Fig. 4a). This MOA$_{NR}$ factor contained several mass peaks associated with amino acids (AA) reported in reference mass spectral signatures of leucine and valine respectively (Schneider et al., 2011). AA signature peaks were identified at m/z 41, 70, 98, 112, 115, 117, 119, 131 and were similar to signature peaks that had been

identified in previous studies by Schmale et al. (2013) in ambient marine aerosols and by Schneider et al. (2011) during a series of laboratory studies on different AA. Similar marker m/z's were present for fatty acid species such as palmitic and oleic acid (Alfarra, 2004). The MOA$_{NR}$ factor had an O/C of 0.53 and an H/C of 1.39, hence much less oxidized than the OOA$_{NR}$ type. These values are intermediate between those often calculated for low volatility, and semi-volatility OOA$_{NR}$ in the ambient atmosphere (Canagaratna et al., 2015), and are similar to

those identified by Schmale et al. (2013) for an amino acid type aerosol (O/C 0.35 and H/C 1.65) detected in an ambient marine aerosol.

The third most prominent factor was identified as a primary organic aerosol ($POA_{NR}$) and contributed by 26% to the total organics at the start of the campaign and decreased to 9% near the end of the campaign (Fig. 4a). This factor contained typical aliphatic signatures and had little contribution from m/z 44. The mass spectral signature of $POA_{NR}$ factor correlated well with reference mass spectra of leucine (r = 0.56) and valine (r = 0.51) but also with fatty acid mass spectra of oleic (r = 0.69), and palmitic acid (r = 0.74). The O/C ratio of this $POA_{NR}$ was 0.1 and the H/C was 1.64, which is typical for values of primary organic aerosol in the ambient atmosphere. The $POA_{NR}$ factor identified in this work, as well as the H/C ratio (1.64), was similar to the aliphatic rich organic aerosol species measured in contained wave chamber experiments during a phytoplankton bloom (Wang et al., 2015). Once the bloom passed the H/C of these aerosol particles decreased, and it was hypothesized that the primary organics were transformed through microbial activity in the water. During the PEACETIME campaign, the $POA_{NR}$ factor had highest concentrations at the start of the field campaign and then later decreased. This decrease in $POA_{NR}$ was accompanied by an increase in the more oxidized $MOA_{NR}$ (Fig. 4b).

The last factor, $MSA-OA_{NR}$, contributed 6% ± 1% and contained typical signature peaks at m/z 65, 79, and 96, and correlated with reference mass spectra of MSA (r = 0.34 (Fig. S8). This mass spectra is similar to that identified by Timonen et al. (2016) in Antarctica. However, unlike previously measured ambient MSA-*like* species (Schmale et al., 2013; Mallet et al., 2019) it contained little or no oxygenated peaks at lower masses (m/z 43, 44, 45) making it impossible to calculate an O/C ratio. As mentioned above, it is possible that, given the low temporal variability, the m/z 44 was incorrectly attributed by the PMF analysis resulting in an excess m/z 44 in OOA and missing m/z 44 in $MSA-OA_{NR}$. The H/C ratio of 1.12 was similar to 1.2 measured by Ovadnevaite et al. (2011), but lower than the reported 1.6 by Schmale et al. (2011) for $MSA-OA_{NR}$, both detected in ambient aerosol. The presence of an MSA-OA in nascent sea spray generated in the present study suggests that this compound is already present in the seawater, and not only produced from gas-phase DMS emissions and oxidation in the atmosphere. The Mediterranean Sea experiences a high level of radiation (Mermex group, 2011), and could also explain the presence of MSA-like compounds from DMS oxidation within the seawater. Precursor species of MSA exists in seawater, such as dimethylsulfoniopropionate. However, the available reference mass spectra of this compound (determined using liquid chromatograph MS) contains m/z values at 63, 73, and 135. (Swan et al., 2017, Spielmeyer and Pohnert, 2010). Therefore we believe that the $MSA-OA_{NR}$ species measured in these seawater samples resembles MSA more than one of its precursor species.

AA containing OA have been measured at a number of coastal sites, and their formation in the ambient atmosphere is similar to that of MSA, where the AA are formed from the gas-phase partitioning of amines such as trimethylamine or dimethylamine into the particulate phase (Facchini et al., 2008). These AA OA signatures were detected during measurements made on the east coast of America, or in the Arctic (Dall'Osto et al, (2019), and also during controlled wave and bubble bursting chamber experiments (Dall'Osto et al., 2019; Decesari et al., 2019, Kuznetsova et al., 2005; Wang et al, 2015).

In our experimental setup, the short time between particle generation and analysis (less than 30 seconds) does not allow for the formation of secondary aerosol through oxidation and partitioning of gas-phase species into the particle phase. Since these amino acid signatures are internally mixed with signatures for several different species, we assume that they are present in the organic matter of the seawater, similar to conclusions made by Dall'Osto et al. (2019).

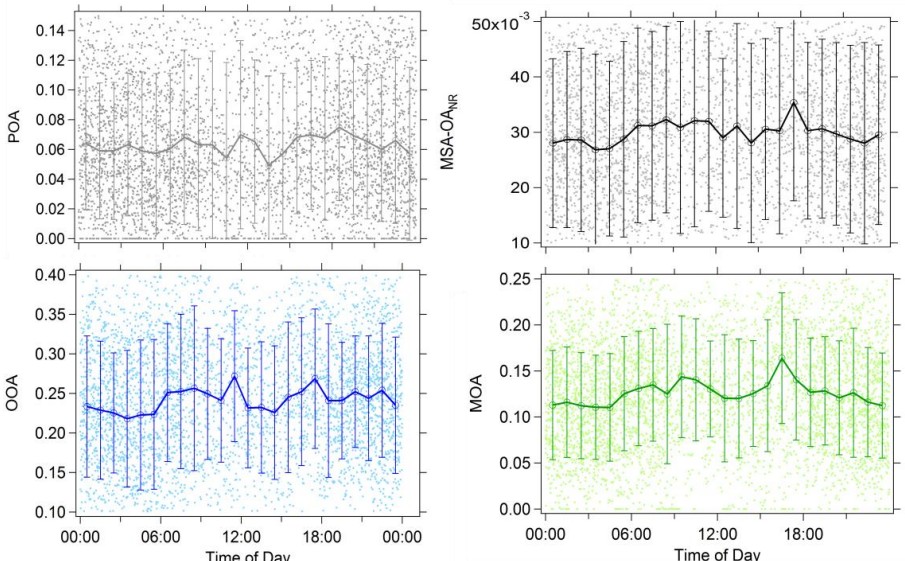

Although very weak and within the measurement error, OOA$_{NR}$, MSA-OA$_{NR}$, and MOA$_{NR}$ factors had a similar diurnal variation with increases during the early hours of the morning and again in the afternoon (Fig 5). POA$_{NR}$ species showed similar variations to the other species in the morning but a second increase was not observed in the afternoon. A lack of strong diurnal variation of MSA-OA$_{NR}$ and OOA$_{NR}$ indicates that the secondary (oxygenated) nature of these compounds might be a result of biological processing rather than photochemical processes.

**Figure 5: Diurnal variation of the four PMF organic factors chosen to represent the measured non refractory organic aerosol: average and standard deviation of the measurements during the whole PEACETIME cruise**

### 3.3. The sources and formation pathways of marine organic aerosol species

In several large-scale climate models *Chl-a* is used as a proxy of phytoplankton biomass to predict the organic fraction of sea spray. In this study, the measured *Chl-a* in the underway surface seawater was low and had little variability ($0.07 \pm 0.013$ mg m$^{-3}$), therefore making it difficult to extract any significant relationship between our measured organic mass fractions and the measured *Chl-a*. No significant correlations were observed between the mass concentrations of the OM (measured by either the ACSM (OM$_{ACSM}$) or on filters (OM$_{filter}$ = WIOM+WSOM) and *Chl-a* concentrations, nor between the fraction of these two organic classes to the total sea spray mass and *Chl-a* (not shown). Therefore it is important to identify other marker species or processes that can be used to correctly link seawater chemical composition, biological activity, and the organic fraction in the seawater aerosol that can represent up to 20% of the total submicron sea spray mass in oligotrophic waters.

In a companion paper, we illustrate that the total number of sea spray particles measured by the DMPS was correlated to the nanophytoplankton cell abundance (NanoPhyto) (r = 0.33, n 501, p < 0.001), (Sellegri et al 2020). The hypothesis behind the dependence of the sea spray number concentration on NanoPhyto is that organic matter released by NanoPhyto influences the surface seawater (SSW) surface tension, and therefore the bubble lifetime that drives the number of film drops ejected to the atmosphere.

### 3.3.1. High time resolution correlations

In this section we will investigate the dependence of each of the organic classes identified in sea spray to the SSW biogeochemical properties. The relationships of the total organic mass concentrations and the fractional organic contributions to the seawater biochemistry were investigated (Fig. 6).

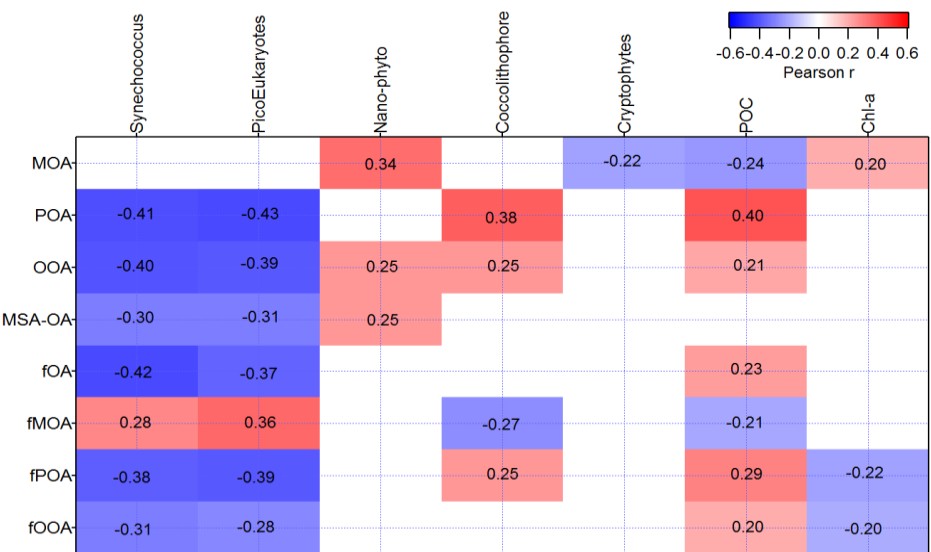

**Figure 6: Pearson correlation matrix showing the agreement of the four different organic factors (MOA$_{NR}$, POA$_{NR}$, OOA$_{NR}$ and MSA-OA$_{NR}$) and their fraction of the total sea spray mass (fMOA, fPOA, fOOA and fOA) with several phytoplankton functional group abundances (cell cm$^{-3}$) (Synechococcus, PicoEukaryotes, NanoEukaryotes (Nano-Phyto), Coccolithophore (Coccolith), Cryptophytes), total _Chl-a_ (mg m$^{-3}$) and POC (mg m$^{-3}$) in the sampled seawater during the whole campaign. Sample number = 461, Correlations with R values < 0.16 had significance values lower than 0.001 and were therefore left blank.**

MOA$_{NR}$ was strongly linked to NanoPhyto (r = 0.34), as was MSA-OA$_{NR}$ and OOA$_{NR}$ but with less significance (r = 0.25). Therefore, these organic classes follow the total sea spray mass and number behavior, as illustrated in Sellegri et al., (2020). The hypothesis of organic matter influencing the surface tension of seawater, bubble lifetime, and the number of film drops, is therefore linked to this specific class of organic matter. Fatty acids, present in the MOA$_{NR}$ and OOA$_{NR}$ spectra, have been reported to be enriched in the SML (Cunliffe et al., 2012), which would explain their impact on the bubble bursting process. POA$_{NR}$ species are instead significantly correlated with particulate organic carbon concentrations. POC (r = 0.40), and Coccolithophore like abundance (r=0.38). A relationship between Coccolithophore-like cell abundance and [POC] is likely linked to the ability of the coccolithophores or similar groups of phytoplankton to secrete large amounts of sticky carbon which can result in the formation of gels and POC (Engel et al., 2004). As the time variation of POA$_{NR}$ does not follow that of total sea spray mass, it is possible that POA is not linked to film drops formation and is ejected into the atmosphere via separate mechanisms, such as jet drops. It was recently shown by Wang et al. (2017) that jet drops can contribute significantly (up to 43%) to the population of submicron SSA. The jet drop-originating SSA has a different chemical composition than the film drop originating SSA, and is more influenced by the SML (Wang et al. 2017). The hypothesis of POA$_{NR}$ being linked to jet drops is backed-up by its relationship to POC in SSW.

The time series of $OOA_{NR}$ had positive relationships with NanoPhyto (r = 0.25), but also with coccolithophore-like cell abundances (r = 0.25) and POC (r = 0.21) and hence $OOA_{NR}$ seem to have an intermediate behavior between $POA_{NR}$ and $MOA_{NR}$. All organic classes except $MOA_{NR}$ are anticorrelated to the small classes of phytoplankton (picoeukaryotes and synechococcus). This anticorrelation could be the result of the competition for nutrients between these small cells and the larger ones that rather drive the POC content.

Except for $MOA_{NR}$, all correlations for the absolute mass of these different types of organic matter are also observed when the fractional contribution of these species to the total mass of SSA (determined from the DMPS) are considered, although less significant (lower part of Fig.6). However, for $MOA_{NR}$ species, correlations with Nano-Phyto no longer hold if the fractional contribution of these species to the total mass of sea spray is considered. Instead, fMOA is correlated to picoeukaryotes and Synechococcus. This is likely due to the strong anticorrelation of the fraction of all other organic classes with these small classes of phytoplankton. At low picoeukaryotes and Synechococcus cell abundances, fPOA, fOOA, and fMSA-OA are higher, artificially decreasing the proportion of fMOA to the rest of the organic matter.

### 3.3.2. Filter-based resolution correlations

Since the non-refractory organic components analyzed using the ACSM technique are only a fraction of the marine organic mass, we investigate the relationships between off-line filter-based organic compounds and seawater biogeochemical properties.

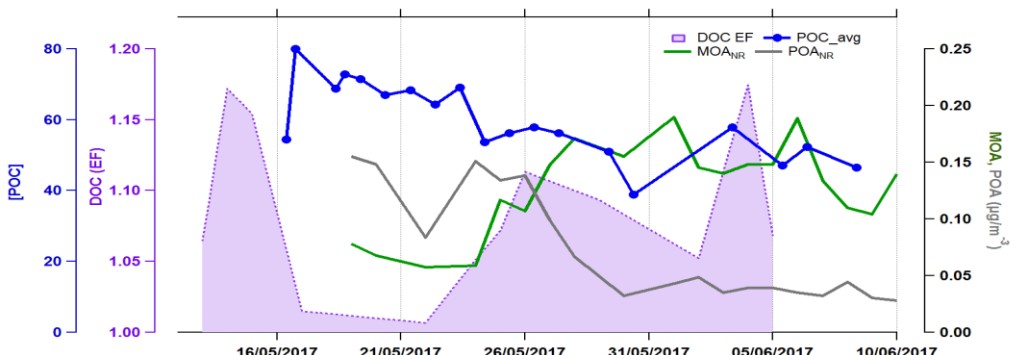

**Figure 7. Time series of the DOC enrichment factor (EF), POC concentrations, and PMF organic factors $MOA_{NR}$ and $POA_{NR}$.**

Filter-based organic fractions are also compared to seawater properties at the filter sampling time resolution. The organic mass concentration from filters is correlated to the coccolithophore cell abundance (r = 0.88, n = 13). The fraction of OM to total mass analyzed on filters (OMSS) was also correlated to coccolithophore cell abundance (r = 0.72, n=13) and POC (r = 0.6, n=13). This indicates that the total organic matter present in sea spray behaved similarly to the non-refractory $POA_{NR}$ and $OOA_{NR}$ analyzed by the ACSM. Previous studies observed a connection between seawater POC and SSA organic fraction. Facchini et al. (2008) found that WIOM in SSA was related to seawater POC derived from microgels. Furthermore, during mesocosm bubbling experiments using *Emiliania huxleyi* cultures and low heterotrophic prokaryote abundance counts, O'Dowd et al., (2015) suggested that the aggregation of DOM into POC in the form of insoluble gel-colloids was the driving

force behind the enrichment of organic matter into submicron SSA. The authors hypothesized that the organic fraction of SSA can be controlled either by DOC or POC, depending on the biological state of the waters.

Measurements of DOC in the SML and underlying seawater were performed daily, as well as surface seawater bacterial total counts from daily surface water samples (5 m depth). DOC was slightly enriched in the SML compared to the underlying seawater, with enrichment factors varying between 1 and 1.2 (Fig. 7). The filter-based total organic content of sea spray aerosol was not correlated to DOC or total bacterial count in the SML, neither was OMSS. However we observe that $MOA_{NR}$ was correlated to the enrichment of DOC in the SML (r = 0.55, n= 9), suggesting again that $MOA_{NR}$ is likely linked to organic matter present in the SML. The correlation between MOA and DOC enrichment in the SML suggests that the fraction of DOC which is enriched in the SML contains lipids and amino acids found in the $MOA_{NR}$ fraction. Although $MOA_{NR}$ is an oxidized organic class, it does not seem to be the result of the bacterial production (BP) of organic matter in the seawater. We actually found a significant anti-correlation between $MOA_{NR}$ and the bacterial production (r = -0.82, n = 10), indicating that $MOA_{NR}$ could instead be consumed by this process.

### 3.3.3. Predicting organic matter in sea spray from seawater biogeochemical properties

By combining relationships between filter-based chemical analysis, ACSM organic source apportionment and seawater properties, it is possible to propose a general relationship that can be used to predict the different fractions of organic matter in the nascent sea spray emitted from oligotrophic seawaters. These different organic fractions may have different atmospheric properties related to their climate impact, such as ice nuclei properties (Trueblood et al, 2020). We chose to parameterize the organic fractions of sea-spray rather than computing organic mass fluxes, for an easier implementation in models that already have an inorganic sea spray source function. However, as shown in the preceding section, the total mass of sea spray is significantly influenced by SSW biology, and we recommend that the biology dependent sea spray number fluxes modulation computed in Sellegri et al. (submitted) is applied before biology-dependent organic fractions are calculated.

The non-refractory organic fraction of nascent sea spray can be predicted with three different equations, with $MSA\text{-}OA_{NR}$ being a negligible fraction of the total sea spray mass:

fPOA = 0.0002 [POC] - 0.001           r = 0.29 , n = 459, p < 0.001           Eq. 2

fOOA= 0.0002 [POC] + 0.02           r = 0.20, n = 478, p < 0.001           Eq. 3

fMOA= $4.5\ 10^{-6}$ x (picoeukaryotes) + 0.009           r = 0.36, n = 459, p < 0.001           Eq. 4

By subtracting all NR organic concentrations (ACSM measured) from the total filter-based organic concentration, we obtain the refractory fraction of the organic matter (RefrOrg). This fraction dominates the organic matter in sea spray. The fraction of RefrOrg to the total mass analyzed on filters (fRefrOrg) is correlated to coccolithophores (r = 0.82, n = 11 p = 0.001) and POC concentrations (r = 0.81, n = 10, p = 0.001), similarly to the $POA_{NR}$ and $OOA_{NR}$ fractions of the sea spray, but with a better correlation to POC. Since POC values are more available than coccolithophore numbers in models or satellite data, fRefraOrg can be computed as follows:

$$fRefrOrg = 0.005 \times [POC] - 0.11 \hspace{4cm} Eq.5$$

These relationships apply to the ranges of POC and picoeukaryotes measured during the PEACETIME cruise. Hence they may be applicable to other oligotrophic waters. If larger ranges of seawater biogeochemical properties are considered in the future, fractions of organic classes should be parameterized as logarithmic laws asymptotic to 1, in order to take into account the saturation of the organic fraction at 1 for the largest POC values.

Figure 8 shows the reconstruction of this dataset using the parametrizations in Eq. 2 to 5. The parametrizations correctly represent the absolute concentrations, with slopes varying from 0.83 up to 1.1. However, the short term variability is less well represented, with correlations varying from 0.20 to 0.49. For this sample size (n = 624) the significance of this correlation is < 0.001. These relatively low correlations suggest that there are parameter, other than POC influencing the emission of these organic

species, especially for OOA that is believed to be linked more to the SML properties than to the bulk seawater and caution should be taken when using these parameterizations for predicting these specific organic classes. However, these parametrizations can be used to provide a first approximation of the organic matter exported to the SSA, and in this type of environment (LNLC)are a better choice than using Chl-*a*.

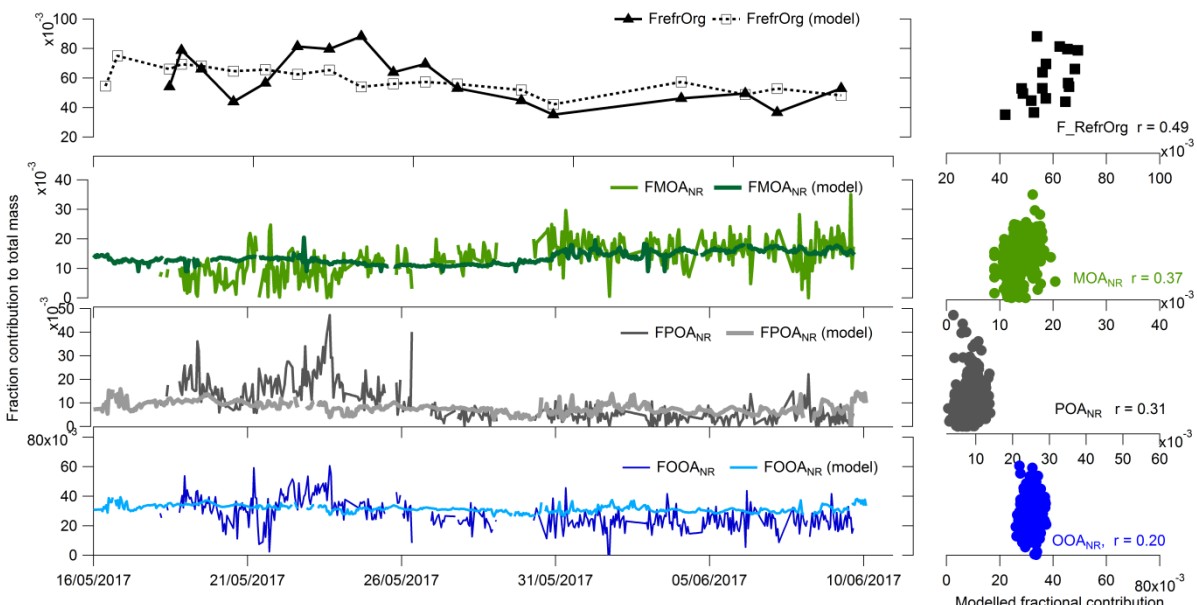

**Figure 8: Time series and correlation plots of each of the parametrizations (Eq 2 to 5) for the determination of the different organic fractions in nascent SSA.**

## 4. Conclusions

The primary objective of this experiment was to study the relationships between sea spray chemical properties

and those of seawater. This work presents a unique dataset, which describes the first deployment of a ToF-ACSM to characterize, in a continuous way, the organic fraction present in sea spray aerosol generated from Mediterranean surface seawater. The non-refractory part of the organic content of sea spray was characterized by

low concentrations and low variability along a 4300 km transect. Yet, using a positive matrix factorization on the ACSM organic mass spectra, it was possible to extract signatures for fatty acids, amino acids, and marine primary organic aerosols in non-refractory nascent sea spray. We identified four organic families: two were composed of mixtures of amino acids and fatty acids (a primary aerosol $POA_{NR}$, and a slightly oxidized $MOA_{NR}$ factor), and two were identified as more oxidized organic aerosol ($OOA_{NR}$ and MSA $OA_{NR}$). The $POA_{NR}$ factor was similar to that observed in wave chamber experiments and correlated well with POC concentrations in the seawater, as did the $OOA_{NR}$ and $MSA$-$OA_{NR.}$. The $MOA_{NR}$ concentrations had a different behavior and correlated well with the nano-phytoplankton cell abundance in the seawater, and also with the total sea spray number concentration and DOC enrichment in the surface microlayer. It is hypothesized that $MOA_{NR}$ is related to surface tension properties that influence the bubble bursting process and the resulting number of film drops ejected to the atmosphere. In contrast, the fraction of $POA_{NR}$, $OOA_{NR}$ and $MSA$-$OA_{NR}$ classes are not connected to the sea spray number concentration, but are linked to POC of the bulk surface seawater and more likely emitted with a different process such as through jet drops.

Off-line chemical analysis of the submicron nascent sea spray provided a general view of the total organic content of these particles, showing that a large part of the organic matter was refractory (to vaporization at 600°C) and thus not detected by the ACSM. However, this refractory organic matter within the nascent sea spray was transferred to the atmospheric aerosol phase similarly to the $POA_{NR}$ concentration found from the ACSM analysis, being significantly correlated to the POC content of the bulk seawater.

This work illustrates the value of continuous aerosol chemistry and physical characterization of the nascent sea spray aerosol in parallel with other biogeochemical measurements in surface seawater. It also illustrates that even under oligotrophic conditions, seawater biogeochemical properties influence the type and concentration of marine organic aerosols and therefore their ability to act as cloud condensation nuclei or ice nuclei. We provide a parameterization of the different marine organic components of nascent sea spray as a function of seawater biogeochemical properties typical for oligotrophic conditions in LNLC regions of the ocean that represents 60% of the global ocean. Such parameterization used in models should allow a better prediction of the impact of living marine organisms on these properties in a future climate.

**Data availability:** Underlying research data are being used by researcher participants of the PEACETIME campaign to prepare other manuscripts, and therefore data are not publicly accessible at the time of publication. Data will be accessible (http://www.obs-vlfr.fr/proof/php/PEACETIME/peacetime.php, last access: 22 June 2020) once the special issue is completed. The policy of the database is detailed here http://www. obs-vlfr.fr/proof/dataconvention.php (last access: 22 June 2020).

**Acknowledgments**

This study is a contribution to the PEACETIME project (http://peacetime-project.org), a joint initiative of the MERMEX and ChArMEx components supported by CNRS-INSU, IFREMER, CEA, and Météo-France as part of the MISTRALS program coordinated by INSU. PEACETIME was endorsed as a process study by GEOTRACES. More information on the PEACETIME cruise can be found at https://doi.org/10.17600/17000300.This work has also received funding from the European Research Council

(ERC) under the European Union's Horizon 2020 research and innovation program (Sea2Cloud grant agreement No 771369). Sea2Cloud was endorsed by SOLAS. We thank the captain and the crew of *the RV Pourquoi Pas ?* for their professionalism and their work at sea.

**Author contributions:** CG and KD designed the PEACETIME project. KS designed the experiments specifically used in this manuscript. KS and AN performed the measurements aboard the ship. MT, GG, NH, JD, IO, F V-W, A.E and BZ were responsible for collecting and analyzing the biogeochemical parameters in either the seawater or in the surface microlayer. MR analyzed the offline filter measurements. EF analyzed the ACSM data with input from LRW and ASHP. EF and KS prepared the manuscript with contributions from all authors.

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
