# Peer review of "Mediterranean nascent sea spray organic aerosol and relationships with seawater biogeochemistry"

_Atmospheric Chemistry and Physics, 2020_

## Referee Comment (RC1) · Anonymous Referee #2 · 17 Jul 2020

**General comments**

The manuscript "Mediterranean nascent sea spray organic aerosol and relationships with seawater biogeochemistry" by Freney et al. presents aerosol composition data collected aboard a 5 week voyage in the Mediterranean Sea in the late-spring - early summer of 2017. The project continuously pumped bulk-seawater from 5m below the surface into a sea-spray generator which uses a plunging jet system to produce primary aerosol. Aerosols generated in the apparatus were then sampled continuously with a range of aerosol instrumentation – of which this paper focuses on the chemical composition measurements: the ToF-ACSM and filter-methods. Surface sea-water was also measured in parallel for phytoplankton cell count, chlorophyll-a, POC. Spo-

radic sampling of the sea surface microlayer was also performed manually, with DOC analysis undertaken on these samples.

Positive Matrix Factorisation was applied to the ACSM mass spectra which identified four significant factors making up the non-refractory organic aerosol (OA) include primary OA, oxidized OA, methanesulfonic acid type OA and mixed OA. Interesting correlations were found between these factors and phytoplankton cell abundance, as well as the POC concentrations. No relationship was observed with chlorophyll-a, but this was likely because of only minimal variance in this parameter throughout the voyage.

The manuscript is well written and provides an interesting dataset to help further understand the organic fraction of primary aerosol generated from bubble bursting. The authors thoroughly assess their data and generally speaking, identify well the limitations of the dataset, however a few conclusions in the discussion are a bit overstated. One of the main limitations of the paper is the model parameterisation section, which leaves much to be desired in terms of convincing the reader that these are robust and significant relationships.

Overall, I would recommend this paper be accepted after minor revisions.

**Specific scientific comments**

- Line 80 requires a reference

- Line 176 – where did the waste-water from rinsing go? Was there any chance it could contaminate or disturb the microlayer before sampling?

- Line 193 – what speed was the air flowing across the water?

- Section 2.4

- – what are the models, manufacturers and setup parameters of the ACSM, DMPS, CPC and impactor?
- – What was the diameter and material of the sampling line? Did you characterise the sample losses?

- Tof-ACSM - did you use a standard or capture vaporizer?

- Table 1 – what is the uncertainty on these? Is it 2 standard deviations, 95

- Section 3.1 – could the discrepancy between the ACSM and filter measurement be more simply explained by the differences in upper diameter limits between the two measurement techniques (i.e. ACSM is 500 nm, presumably the filter is 1um)?

- I would like to see a graph of the size distributions, even in the supplementary information.

- Line 336 – please include the reference for the "reference mass spectra"

- Figure 4 – what are the 3 different pie graphs? Please label or include in the caption.

- Figure 5 and its conclusions – the error bars on this figure are very difficult to read, please consider offsetting them or altering the presentation here. Looking at this figure, complete with the error bars, I find it difficult to come to the conclusions the authors have about there being a diurnal cycle present at all. There is variability from hour to hour, but given the size of the standard deviation, none of this is significant enough to conclude any diurnal cycle.

- While I commend the authors on their efforts to parameterise the production of organic aerosol from POC concentrations, I would like to see the actual underlying data supporting this from your study – i.e. the figures showing POC vs the

fractions of OA, and how your models actually fit the data. I wouldn't put a lot of confidence in the equations without seeing the data, and even then, the Pearson coefficients are reasonably low so trusting and implementing this may be a stretch.

**Technical comments**

- Line 98 – name of vessel should be italicised. Does it need the "?"?

- Line 196 – "different" should be "various"

- Line 312 – remove space after "1.2 g cm-3"

- Line 313 – missing close parenthesis in sentence.

- Figure 3 caption - please check units

- Line 326 – remove full-stop before "(Fig S7)".

- Line 428 – remove semicolon in "Fig; 6"

---

## Referee Comment (RC2) · Anonymous Referee #1 · 21 Jul 2020

The manuscript presents chemical speciation measurements from the oligotrophic Mediterranean Sea water bubble bursting experiments and the source apportionment of marine organic matter (OM). It aims at relating the marine organic concentration and speciation to sea water composition and presents parameterisations for different types of organics. This is potentially very valuable data set, but lacks some better analysis and discussion. Three major findings like refractory OM, OM source apportionment and finally parameterisation are little convincing as detailed below.

Major comments: The presence of refractory OM is not proven, but just speculated. The evidence in the paper can alternatively point to some problems with the ACSM instrument or, more likely, to potential instrument overload with the sea salt. Until this is proven incorrect, the refractory OM origin is little convincing. Especially that

[Figure]

the missing fraction is increasing with time and/or the sea salt load. High sea spray concentrations exceeding 30 $\mu$g/m3 are not representative of ambient sea spray and may well overload the ACSM heater, making measurements non quantitative, thus, causing the registered disagreement with the filters. Was there any relation to RH? Why there is such variation in the sea salt concentrations if these are not ambient, but bubble bursting experiments?

Moreover, an origin of sulphate in the ACSM measurements is not discussed, is it sea salt SO4? Why there is a variation in SO4 fraction with measurement time if it supposedly comes from the sea spray with constant SO4 fraction? Also, why such a big difference between filter SO4 and ACSM measured, refractory? Discussion is required to explain these issues.

Figure 2 and Figure 3 clearly show that the missing fraction goes up with time and after the high total loadings (grey goes up just after the black goes down). The agreement with DMPS deteriorates after high sea salt loading. Pointing again to overload rather than refractory OM. Also the feature observed and discussed in lines 323-324 might well be due to overload rather than the theory presented that would in any case need more discussion and evidence than it is presented now. How the OM discrepancy between filters and ACSM changes with time, is it like one with DMPS for which agreement is worse after high sea salt loads?

Secondly, source apportionment section lacks any detail and objective factor selection information and justification. Why 4 factor solution and not 3 or 5 or, maybe, 6? Correlations with the external reference profiles are very weak and do not exactly justify the factor selection. But even that (correlations as a base for factor selection) is only stated in the text, but not described at all. Factors as they are now might be a result of factor splitting (the lack of oxygenated m/z in some profiles, like MSA, but appearing in much higher contributions in others, like OOA). What is the justification for splitting MOA and POA? They both correlate well with leucine and valine, could they be from the same source? Discussion around this point and quantitative statistical reasons

of separating them is lacking. OOA correlation with reference profiles is neither presented nor discussed. High O/C ratio observed in OOA might well be due to wrongly separated factor. What is the justification for removing sea salt ions, is the instrument mass resolution high enough to assure that only sea salt ions were removed? If it is unit mass resolution, you have removed the associated organic contribution. Wouldn't it be better to have these in the PMF and separate a sea salt factor with corresponding explanation of its origin? Justify the removal of m/z 29 as opposed to downscaling.

Moreover, if MSA was indeed produced by bubble bursting, this could be a significant result, but, again, lacks comprehensive analysis and discussion. Could it be that you have detected some MSA precursors in the water like dimethylsulfonium propionate? or something else? What is the strong evidence for MSA presence? Poor MSA correlation (<0.2) to external profile is very unconvincing for this factor selection in this particular PMF solution. Is 0.2 even significant? Why a diurnal MSA trend was not observed if it was indeed from water oxidation? It is not even discussed along other diurnal trends in Figure 5, why? Discuss the possibility of oxygenated MSA m/z being wrongly attributed to OOA and the consequences on the MSA factor and the overall PMF results.

Finally, the parametrisation of OM was not shown in detail and lacks validation to be considered by a modelling community. The reproduction of the measurements in the manuscript is not shown. What fraction of variability is explained? Fractions of OM are not shown anywhere in the manuscript, just mentioned briefly in this paragraph.

The overview of previous chamber experiments related to marine aerosol in the Introduction is quite shallow, I agree that there are only few studies to date, but even these few are missing in the Introduction.

Is the standard CE applicable here? Provide assumptions/basis for this (lines 271-272).

The introduction of jet drops at lines 445-447 is questionable. Needs better discussion. Would jet drops not produce larger particles than those that are measured with PM1 instruments, both online and offline? Also, the different POA time variation could occur

due to sea salt replacement by primary organics in PM1, which would cause these different sea salt and OM time trends (higher OM enrichment or fraction would mean less sea salt in the sea spray, where higher OM result in lower sea salt while overall sea spray concentration remains the same).

Discussion at lines 328-330 is not elaborate enough. To me, the WSOM shows a very different trend to OM from ACSM. What causes the ACSM peak around 21-25 May, which is completely missed by the filter analysis? The difference in time trends makes me question the fact that online and offline techniques were measuring the same thing, can you at least show one compound that has a good agreement between the two techniques? Why WIOM has fewer points than WSOM? Explain this in the manuscript.

Specific comments: Abstract states 15 min time resolution or ACSM while it is 10 min in the text (line 244). Be consistent, and state what was the real time resolution.

Provide reference for statement on lines 49-50 on the sea spray emissions

Fig 1: provide units for Chl-a and POC

Line 147: what was the time resolution for these measurements? Also, for other measurements of water composition like Synechococcus, PicoEukaryotes, NanoEukaryotes, Coccolithophore, Cryptophytes, etc.?

Lines 225-226: provide more details on acidification before analysis or provide an appropriate reference to the method

Lines 287-288: correct the sentence, it has 'showed' and 'is' in the same sentence

Line 312: why the real sea salt density was not used for the sea salt, which is 2.165 rather than 1.75. What other inorganics were there?

Figure 4c: correct the labelling, it is either shifted or labels are given to minor m/z, difficult to read this figure.

Line 371: provide the H/C value

Lines 414-415: I'm confused with this statement, do you refer to satellite based or concurrently measured Chl-a or both?

Line 415: I question the use of appropriate reference here as, to my knowledge, Rinaldi et al. 2013 did not discuss these issues. Provide the correct reference.

---

## Author Comment (AC1) · 1 Dec 2020

The authors wish to thank the two reviewers for their constructive comments that have helped to improve the quality of this manuscript. Our response to each of the reviewers comments is included in the attached document.

Please also note the supplement to this comment:
https://acp.copernicus.org/preprints/acp-2020-406/acp-2020-406-AC1-supplement.pdf

---

## Author Response (AR1)

The authors wish to thank the two reviewers for their constructive comments that have helped to improve the quality of this manuscript. Our response to each of the reviewers comments is in blue text below. Text added to the manuscript is in quotation marks

**Anonymous Referee #1**

The manuscript presents chemical speciation measurements from the oligotrophic Mediterranean Sea water bubble bursting experiments and the source apportionment of marine organic matter (OM). It aims at relating the marine organic concentration and speciation to sea water composition and presents parameterizations for different types of organics. This is potentially a very valuable data set, but lacks some better analysis and discussion. Three major findings like refractory OM, OM source apportionment and finally parameterization are little convincing as detailed below.

Major comments:
1.The presence of refractory OM is not proven, but just speculated. The evidence in the paper can alternatively point to some problems with the ACSM instrument or, more likely, to potential instrument overload with the sea salt.

With the exception of a relatively high mass loading (25 μg/m$^3$) at the start of the sampling period, the total ACSM mass loading never exceeded 25 μg/m$^3$ (average values were 10 ± 3.4 μg/m$^3$), filter mass loadings averaged 20 ± 8 μg/m$^3$, with highest concentrations measured at the start of the campaign. This mass loading is higher than previous studies of ambient marine aerosol (Coe et al, 2006, Ovadnevite et al., 2012, 2017, Schmale et al., 2013), but is well within the aerosol loadings that can be measured by these instruments (up to 100's of ug/m3). One potential issue with high mass loadings of slowly vaporizing species, is that the background (filter) measurement changes slowly and causes an artefact in the sample-filter measurement of the particle signal. For example, this has been observed to lead to "negative" Cl measurements in a Q-ACSM (Tobler et al., AMT, 2020). We can rule out this issue in these measurements for two reasons. First, the timing was 2 minutes in filter and 8 minutes in sample, giving more than adequate time for the background to adjust to the change from filter to sample and back. Second, a series of particle-free sampling periods were performed throughout the campaign, during which time the backgrounds decreased, but not by an amount that would suggest overloading the instrument (Fig R1).

[Figure]

Figure R1. Showing SO4 and Org all (open+closed signals) and a particle free sampling period (within the dashed box area) at a) the start of the campaign and b) near the end of the campaign.

2.Until this is proven incorrect, the refractory OM origin is little convincing. Especially that the missing fraction is increasing with time and/or the sea salt load.

As illustrated in Figure 2, and again in Figure 3a, the missing fraction occurs during some short periods at the start of the sampling and then consistently at the end of sampling. However, during this time the fraction of missing mass did not vary with increasing mass (Fig. R2).

[Figure]

Fig. R2 Ratio of the missing mass to the total measured mass

3. High sea spray concentrations exceeding 30 μg/m3 are not representative of ambient sea spray and may well overload the ACSM heater, making measurements non quantitative, thus, causing the registered disagreement with the filters. Was there any relation to RH?

Authors: Although the concentrations that are measured here are higher than ambient marine measurements, we do not believe that overloading is an issue. This is illustrated in our response to the previous questions. The RH fluctuated between 20 and 45% (Fig.S1). This graph is now included in the supplementary figures. No trend was observed in the difference between DMPS volume and ACSM volume as a function of RH.

[Figure]

.

Figure S1: The relative humidity (Blue), The ACSM volume concentration (green) and the DMPS total volume concentration as a function of time.

4. Why there is such variation in the sea salt concentrations if these are not ambient, but bubble bursting experiments?

We have added text into the manuscript:

Page 11, Line 348: "The variation in the total seaspray mass concentration with time is a result of the variability in the number of seaspray particles generated from the sea spray generation device. The size distribution of the aerosol remained very stable throughout the experiment, but the seaspray number emission flux is influenced by the variability in seawater biogeochemical properties. This is presented and discussed in detail in a companion paper (Sellegri et al., 2020), where we show a clear relationship between the sea spray number concentrations generated in the bubble bursting system and the nanophytoplankton concentration of the seawater, not only in the PEACTIME experiment but also in other seawater types.

Also, why such a big difference between filter SO4 and ACSM measured, refractory? Discussion is required to explain these issues. Moreover, an origin of sulphate in the ACSM measurements is not discussed, is it sea salt SO4?

More discussion is added to the text. We initially did not alter the RIE of $SO_4$ in our instrument since we did not have a calibrated RIE value for $ssSO_4$. The RIE value used for $SO_4$ is that determined for $(NH_4)_2SO_4$.

Considering the reviewers comments, and assuming that no other sources of $SO_4^{2-}$ could be present in the sample volume, we adjusted the RIE of the $SO_4^{2-}$ measured by the ACSM so that a good comparison with filter measurements was obtained. A $SO_4^{2-}$ RIE of 0.3 resulted in a relatively good agreement with the filters (Fig.R3). Using an RIE of 0.3 for the $SO_4^{2}$, we observe that the $SO_4^{2-}$ from the ACSM follows relatively well that of the filters for the start of the field campaign, and is lower than the filter $SO_4$ near the end of the campaign, similar to the trend for total mass in Fig. S3.

Additional text:

Line 272: "However, considering that the measured $SO_4$ concentrations represent those of seasalt(ss) $SO_4$, we adjusted the RIE of $SO_4$ to 0.3. Using this value we observe that the mass concentration of $SO_4^{2-}$ from the ACSM

follows that of the filters for the start of the field campaign, but is lower than the filter $SO_4$ concentration near the end of the campaign".

Line 227: "The contributions of the estimated $ssSO4$ (seasalt $SO_4$), ssK, ssMg, ssCa, and ssCl were calculated based on the seawater theoretical ratio (Seinfeld and Pandis, 2006 (these values are: $SO_4$: 0.25, K: 0.06, Mg: 0.12, Ca:0.04)). The remaining nss fraction of the inorganic aerosol was within the measurement error of the instrument. "

[Figure]

Figure R3. Comparison of $SO_4$ measured by the ACSM (using an RIE value of 0.3) and that measured on filters.

6. Why there is a variation in $SO_4$ fraction with measurement time if it supposedly comes from the sea spray with constant $SO_4$ fraction?
In the primary sea salt emissions, it is possible that the fraction of $SO_4$ compounds can vary over time depending on the associated compounds (e.g Mg, $Ca^+$) in the sea salt aerosols. In addition, these variations are also a result of the ACSM measuring less efficiently the refractory mass concentration compared with the filters.

Additional text:
Page 10 Line 339 "The variability in the different chemical compositions throughout the sampling is thought to be a result of the differing associated contributions of refractory compounds (Ca, Mg, K etc) in the sea salt sample."

7.      Figure 2 and Figure 3 clearly show that the missing fraction goes up with time and after the high total loadings (grey goes up just after the black goes down). The agreement with DMPS deteriorates after high sea salt loading.  Pointing again to overload rather than refractory OM.

The missing mass is starting to increase exactly during the night of May 31 to June 1st, right after a period of low total loading. There is no evidence of overloading of the instrument, nor of any changes to the instruments turning parameters.  This period corresponds to a change of seawater type when more inputs from the Atlantic Ocean occurs and a corresponding decrease in the seawater salinity is observed (Fig S11).

Additional text and figures are included in the manuscript
Line 361: "The later period of the field campaign corresponds to a change in seawater type with more inputs from the Atlantic Ocean occurs, and where a corresponding decrease in practical seawater salinity is observed (Fig. S11). It is possible that during this sampling period the seawater contains higher fractions of refractory

material that are less efficiently measured by the ACSM. A second way to estimate the mass missed by the ACSM is to subtract the total ACSM mass loading from the total mass measured on the filters. Comparing this estimate of missing mass to the different species measured on the filters shows the best correlations with $Mg^{2+}$, $Ca^{2+}$, and $SO_4^{2-}$ (Fig. S12). This might suggest that the ability of the ACSM to measure NaCl particles depends on how NaCl is associated with other compounds in the sea spray."

[Figure]

Figure S11: The practical salinity measurement of the seawater as a function of time.

8. Also the feature observed and discussed in lines 323-324 might well be due to overload rather than the theory presented that would in any case need more discussion and evidence than it is presented now. How the OM discrepancy between filters and ACSM changes with time, is it like one with DMPS for which agreement is worse after high sea salt loads?

The reviewer addresses the sentence

*"might suggest that the ability of the ACSM to measure NaCl particles depends on how NaCl is associated with other compounds in the sea spray."*

Our responses to previous comments explain why overloading the ACSM with salt is not an issue. The paragraph on the OM comparison has been reworded.

Line 371:"The PM1 mass concentrations of OM calculated from the filters were additionally compared with the total OM measured from the ACSM ($OM_{ACSM}$) (Fig. S13). The $OM_{ACSM}$ represented on average 27% of the total filter OM, 40% WIOM, and 68% WSOM. The differences observed between the filter and $OM_{ACSM}$ measurements are likely OM particles internally mixed with the unsampled Refr particle, and hence not measured by the ACSM."

[Figure]

Figure S13: Comparison of organic aerosol measurement by the ACSM with water soluble and water insoluble organic matter (WSOM, WIOM) measured from filters.

9.      Secondly, source apportionment section lacks any detail and objective factor selection information and justification. Why 4 factor solution and not 3 or 5 or, maybe, 6? Correlations with the external reference profiles are very weak and do not exactly justify the factor selection. But even that (correlations as a base for factor selection) is only stated in the text, but not described at all.

More detailed discussion is now included in the supplementary material including the three and five factor profiles and times series as well as their respective correlations with reference spectra.

In the supplementary material.

"Our choice of factor solution was dependent on a number of different variables, including mass spectral signatures, correlations with reference mass spectra, and time series variability. This data set had very low temporal variability making it difficult to separate species based on the last factor alone. This is also reflected in the small change in average $Q/Q_{exp}$ values across three, four, and five factor solution (see Fig. S4). The PMF analysis was capable of describing 70% of the total organic mass, with the remaining 30% being unexplained noise. The fractional contibution of each of the organic factors as a function of time is illustrated in Fig.S5 for the four factor solution.

[Figure]

**Figure S4. The $Q/Q_{exp}$ results for 3, 4 and 5 factor solutions. The four factor solution (used in the discussion) is highlighted in blue. An f-peak value of 0 was chosen (yellow circle).**

[Figure]

Figure S5. Fractional contribution of factors as a function of time.

In the two factor solution, an oxidized organic aerosol (OOA) and a second less oxidized species was identified. In the three factor solution, the OOA mass spectral profile remained unchanged and was typical of reference

mass spectra for LVOOA (Fig. S6, S7). The less oxidized OA separated into what we refer to as a primary OA (POA$_{NR}$) and the marine OA (MOA$_{NR}$). The POA$_{NR}$ had signature peaks for primary organics m/z 55, 57 but also peaks at mz 69, 71, 91 correlating with mass spectral profiles of proline, valine, palmitic acid and typical hydrocarbon OA (Fig. S7). The MOA$_{NR}$ had signature peaks at m/z 41, 70, 98, 112, 115,, 117, 119, 131, and similar to the POA species, correlated with reference mass spectra of leucine, proline, and valine, but also with more oxidized reference mass spectra of oxalic acid and malonic acid. It did not contain signature peaks of hydrocarbon species like POA. POA$_{NR}$ species were measured in higher concentrations at the start of the sampling with the MOA$_{NR}$ taking over near the end of the campaign. This pattern remained consistent in the 3, 4, and 5 factor solutions (Fig. S6, Fig. 4b in the main text and Fig. S9).

[Figure]

**Figure S6: Factor profiles and time-series for the three factor solution.**

[Figure]

**Figure S7: Correlation of the three factor solution profiles with a number of reference mass spectral profiles.**

As described in the text a four factor solution was chosen for this data set. The four factor solution included a methanesulphonic (MSA) acid-like OA in addition to the previously identified $MOA_{NR}$, $POA_{NR}$, and $OOA_{NR}$ (Fig 4, main text). The MSA-like factor only correlated with the reference mass spectra for MSA with a value of r=0.34) (Fig S8).

[Figure]

**Figure S8: A four factor solution and the corresponding mass spectral profile correlations.**

When extending the mass spectral solution to a five factor solution, each of the previous factor profiles were preserved but the MOA$_{NR}$ split, giving an additional species with similar temporal and mass spectral profiles (Fig. S8).

[Figure]

**Figure S9: A five factor solution.**

[Figure]

**Figures S10 Correlation matrix of five factor correlation."**

10.  Factors as they are now might be a result of factor splitting (the lack of oxygenated m/z in some profiles, like MSA, but appearing in much higher contributions in others, like OOA).

More details have been included in the discussion of the PMF analysis and in the supplementary material (see response to previous comment).

Main text: Line 383: "A more detailed discussion of the solutions from 3 to 5 factors is provided in the supplementary material as well as correlations with external mass spectral and time series. Based on the on correlations with reference mass spectra, and on observations of the temporal variations we chose a four factor solution."

Line 398 "It is also possible that the PMF analysis wrongly attributed excess m/z44 to OOA at the expense of other species such as the MSA-OA$_{NR}$ discussed below. This would impact the reported absolute concentrations of the OOA$_{NR}$ vs the MSA-OA$_{NR}$."

Line 434: "As mentioned above, it is possible that, given the low temporal variability, the m/z 44 was incorrectly attributed by the PMF analysis resulting in an excess m/z 44 in OOA and missing m/z 44 in MSA-OA$_{NR}$"

11. What is the justification for splitting MOA and POA? They both correlate well with leucine and valine, could they be from the same source? Discussion around this point and quantitative statistical reasons of separating them is lacking.

We have added more details into the main text (listen above in previous response), and in the supplementary material (listed below). We hope that this additional information provides sufficient details to justify our choice in the factor solutions

12. OOA correlation with reference profiles is neither presented nor discussed. High O/C ratio observed in OOA might well be due to wrongly separated factor.

We added in several more reference mass spectral profiles to compare with the OOA factor. This information is now included in both the supplementary text and the main text. Details are above in previous responses.

13. What is the justification for removing sea salt ions, is the instrument mass resolution high enough to assure that only sea salt ions were removed? If it is unit mass resolution, you have removed the associated organic contribution.

The TOF-ACSM gives unit mass resolution (UMR) mass spectra, so we cannot distinguish between salt and non-salt ions at a given m/z. It is correct that removing the m/z's associated with NaCl (23, 35, 37, 58, 60, 81 and 83) will also remove contributions from organics at these m/z's. However, the organics are typically a factor of 10 smaller than sea salt in the ACSM signal, so that the organic contribution at these m/z's is very small and difficult to estimate. We did run PMF on the full set of m/z's and found that these m/z's were entirely attributed to sea salt factors, so PMF is unable to identify the organic contributions. If we try to use the fragmentation table to calculate the organic contribution at these m/z's from other organic m/z's, this would not add any new temporal information for PMF to analyze. This justifies removing these m/z's from the organic mass spectral input to PMF.

Additional text:
Line 291: "The TOF-ACSM gives unit mass resolution (UMR) mass spectra, so it is not possible to distinguish between salt and non-salt ions at a given m/z. A decision was made to remove all sea salt related ions from the organic mass spectral data matrix, giving a total of 116 m/z from 0 up to 150. We are aware that removing the m/z's associated with NaCl (23, 35, 37, 58, 60, 81 and 83) will also remove contributions from organics at these m/z's. However, the organic contributions at these m/z values are small relative to the rest of the organic MS and are typically a factor of 10 smaller than sea salt in the ACSM signal

13. Wouldn't it be better to have these in the PMF and separate a sea salt factor with corresponding explanation of its origin? Justify the removal of m/z 29 as opposed to downscaling.

As mentioned above, we did run PMF on all m/z's. However, because the organic signal is small compared to the sea salt and because the organics do not vary much in time, this yielded solutions with many sea salt/other inorganic containing factors that tended to split before organic factors could be resolved. By removing the sea

salt and other inorganic contributions, we were able to obtain PMF solutions for the organics that had a reasonable number of factors with identifiable mass spectral patterns.

The justification for removing m/z 29 is as follows. The ToF-ACSM was run with 2min filter/8 min sample. For m/z's that have a high background from air, such as 29 from 15NN, and when trying to analyze such low signals with long times between filters, any drift in sensitivity can often result in an apparent difference signal that is an artefact. Therefore, when leaving m/z 29 in, PMF identified a factor that was predominantly m/z 29, but the time series was very noisy and flat. Downweighting m/z 29 (by x100) did not help distribute the signal at m/z 29 into the other organic factors. Since m/z 29 was only contributing to noise and not to chemical information, we decided to remove it.

Additional text: Line 298:

"In addition, during sampling the ToF-ACSM was run with a 2 min filter/8 minute sample. When sampling with long times between filters, any drift in sensitivity can result in a difference signal that is an artefact. This is especially try for those signals with a high background (e.g. from air at m/z 29 from $^{15}$NN), and when measuring low concentrations of organic mass. In PMF solutions with the m/z 29, the factor contained predominantly m/z 29, and the time series was noisy and flat. Downweighting m/z 29 (by x100) did not help distribute the signal at m/z 29 into the other organic factors. Therefore since m/z 29 was only contributing to noise and not to chemical information, we decided to remove it."

15.     Moreover, if MSA was indeed produced by bubble bursting, this could be a significant result, but, again, lacks comprehensive analysis and discussion. Could it be that you have detected some MSA precursors in the water like dimethylsulfonium propionate? Or something else?

There is currently no reference mass spectrum for dimethylsulfoniopropionate on the NIST website, but a number of publications show that the mass spectral profiles of DMSP determined with LC-MS, have dominant contributions from m/z 63, 73, and 135. We added these references and a short discussion of this possibility in the discussion.

 Line 434   "Precursors of MSA could exist in seawater, such as dimethylsulfonium proprionate. However, the available mass spectral signatures (determined using liquid chromatography-MS) show m/z values at 63, 73, and 135 (Swan et al., 2017, Spielmeyer and Pohnert, 2010). Therefore we believe that the MSA-OA$_{NR}$ species measured in these seawater samples resembles MSA more than one of its precursor species."

Swan, H.B., Deschaseaux, E.S.M., Jones, G.B. *et al.* Quantification of dimethylsulfoniopropionate (DMSP) in *Acropora* spp. of reef-building coral using mass spectrometry with deuterated internal standard. *Anal Bioanal Chem* 409, 1929–1942 (2017). https://doi.org/10.1007/s00216-016-0141-5

Spielmeyer, Astrid and G. Pohnert. "Direct quantification of dimethylsulfoniopropionate (DMSP) with

hydrophilic interaction liquid chromatography/mass spectrometry." *Journal of chromatography. B, Analytical technologies in the biomedical and life sciences* 878 31 (2010): 3238-42 .

16.     What is the strong evidence for MSA presence? Poor MSA correlation (<0.2) to external profile is very unconvincing for this factor selection in this particular PMF solution. Is 0.2 even significant?

The only evidence that we have for MSA is the presence of the typical MS peaks that correspond to the reference mass spectra. There are 116 peaks in each MS, therefore a correlation of 0.34 has a significance of < 0.05. The missing $m/z$ 44 leads to a poor correlation .We have taken into consideration the reviewers comments and integrated these into the discussion.

17.  Why a diurnal MSA trend was not observed if it was indeed from water oxidation? It is not even discussed along other diurnal trends in Figure 5, why?

Although very weak and within the error, MOA, OOA, and MSA had very similar diurnal variations with a peak in the early morning at around 10 am and a second increase in the afternoon at around 16hrs. Lowest concentrations are measured between 12 and 14hrs. POA variations increased similar to the other species in the morning but showed less variation throughout a 24 hr period. A lack of strong diurnal variation of MSA (and OOA) indicates that the secondary (oxygenated) nature of these compounds might be due to biological processing rather than photochemical processes.

Line 451: "Although very weak and within the measurement error, $OOA_{NR}$ $MSA-OA_{NR}$, and $MOA_{NR}$ factors had a similar diurnal variation with increases during the early hours of the morning and again in the afternoon (Fig 5). $POA_{NR}$ species showed similar variations to the other species in the morning but a second increase was not observed in the afternoon. A lack of strong diurnal variation of MSA and OOA indicates that the secondary (oxygenated) nature of these compounds might be primarily a result of biological processing rather than photochemical processes. "

18.     Discuss the possibility of oxygenated MSA $m/z$ being wrongly attributed to OOA and the consequences on the MSA factor and the overall PMF results.

The wrong attribution of oxygenated MSA being attributed to OOA could potentially mean that the identified MSA is largely underestimated.

19. Finally, the parametrization of OM was not shown in detail and lacks validation to be considered by a modelling community. The reproduction of the measurements in the manuscript is not shown. What fraction of variability is explained?

Text and figures are included in the text

Line 579: "Figure 8 shows the reconstruction of this dataset using the parameterizations in Eqs 2 to 5. The parametrizations correctly represent the absolute concentrations, with slopes varying from 0.83 to 1.1. However,

the short term variability is less well represented, with correlations varying from 0.20 to 0.49. For this sample size (n = 624) the significance of this correlation is < 0.001. These relatively low correlations suggest that there are parameters other than POC influencing the emission of these organic species, especially for OOA that is believed to be linked more to the SML properties than to the bulk seawater and caution should be taken when using these parameterizations for predicting these specific organic classes. However, these parametrizations can be used to provide a first approximation of the organic matter exported to the SSA, and in this type of environment (LNLC) are a better choice than using Chl-*a*.

[Figure]

Figure 8: Time series and correlation plots of each of the parametrizations (Eq 2 to 5) for the determination of the different organic fractions in nascent SSA."

20. Fractions of OM are,not shown anywhere in the manuscript, just mentioned briefly in this paragraph.

**Fractions of each of the resolved OM species as a function of time are now included in the supplementary in Fig. S5.**

[Figure]

21.    The overview of previous chamber experiments related to marine aerosol in the Introduction is quite shallow, I agree that there are only few studies to date, but even these few are missing in the Introduction.

A number of additional references have been included (in red). However, if there are some other relevant references that we may have omitted we would be happy to include them.

A limited number of studies have focused directly on the composition of nascent sea spray aerosol particles emitted from wave action in controlled simulation chambers (Wang et al., 2015, Collins et al., 2016), or through dedicated bubble bursting experiments in ambient environments (Bates et al., 2012; 2020, Dall'Osto et al., 2019, Park et al., 2019). These studies in controlled environments identified the presence of aliphatic-rich and amino acid-rich organic aerosols related to different phases of phytoplankton blooms (Bates et al. 2012; Wang et al., 2015). Dall'Osto et al., (2019) identified an amino acid contribution in both nascent sea spray aerosol and ambient aerosols. Park et al.,(2019) observed that sea-salt aerosol production was positively correlated with organic compounds in the water, notably dissolved organic carbon, marine microgels and chloropyll a (Chl-a). However, in all of these studies, samples were collected at point intervals and were not able to provide information on the evolution aerosol physical and chemical properties over a large spatial area In addition to this, previous studies do not provide measurements making it possible to provide the quantitative link between seawater biogeochemistry and the SSA organic composition.

Is the standard CE applicable here? Provide assumptions/basis for this (lines 271-272).
The collection efficiency of refractory aerosol has been shown to be much lower than non-refractory species. However, mixtures of refractory and non-refractory compounds can have a CE closer to the default value of 0.5 for typical ambient measurements. Applying a standard CE of 0.5 still provided us with a relatively good agreement with the DMPS volume for salt and SO4 during the first part of the campaign. For this reason, and without having a validated reason for applying a different CE we stayed with 0.5

22.    The introduction of jet drops at lines 445-447 is questionable. Needs better discussion. Would jet drops not produce larger particles than those that are measured with PM1 instruments, both online and offline?

The reviewer refers to the following statement in the manuscript: "As the time variation of $POA_{NR}$ does not follow that of total sea spray mass, it is possible that POA is not linked to film drops formation and is ejected into the atmosphere via separate mechanisms (such as jet drops)."

We now discuss this hypothesis in more details, and also include the recent publication by Wang et al. (2017) that states that jet drops can contribute significantly (up to 45%) to the population of sub micron SSA. The text now reads:

Line 499 : As the time variation of $POA_{NR}$ does not follow that of total sea spray mass, it is possible that POA is not linked to film drops formation and is ejected into the atmosphere via separate mechanisms (such as jet drops). It was recently shown by Wang et al. (2017) that jet drops can contribute significantly (up to 43%) to the

population of submicron SSA. The jet drop-originating SSA has a different chemical composition than the film drop originating SSA, and is more influenced by the SML (Wang et al. 2017). The hypothesis of POANR being linked to jet drops is backed-up by its relationship to POC in SSW.

23. Also, the different POA time variation could occur due to sea salt replacement by primary organics in PM1, which would cause these different sea salt and OM time trends (higher OM enrichment or fraction would mean less sea salt in the sea spray, where higher OM result in lower sea salt while overall sea spray concentration remains the same).

The POA concentrations represent a very small fraction of the total mass compared to that of NaCl, so it is unlikely that we would be able to observe this change (Figure R4).

[Figure]

Figure R4. Fraction of POA vs Fraction of Seasalt measured over the entire campaign.

24. Discussion at lines 328-330 is not elaborate enough. To me, the WSOM shows a very different trend to OM from ACSM. What causes the ACSM peak around 21-25 May, which is completely missed by the filter analysis? The difference in time trends makes me question the fact that online and offline techniques were measuring the same thing,,can you at least show one compound that has a good agreement between the two techniques?

The peak on 23/05/2017 can be attributed to higher variability in the ACSM measurements during this period, and to a contamination peak (opening up the sampling bath) that had not be removed prior to averaging. There was a period of high concentration that should have been filtered out and was included in the averaging of the sampling period. These data were recalculated (removing any flagged spikes in concentration). The update figure S13 is shown above in response to comment no. 7.

25a: Why WIOM has fewer points than WSOM? Explain this in the manuscript.
More details are included in the main text as well as detection limits for both OC and WSOC measurements.

More text is included into the main text to describe this:
Line 233:The water-soluble organic carbon (WSOC) content of the extracts was quantified using a total organic carbon (TOC) thermal combustion analyzer (Shimadzu TOC-5000A) (Detection limit (DL) = 1.9 µgC/filter). Measurements of the total carbon (TC) content were performed on a filter punch cut before water extraction by a thermal combustion analyzer equipped with a furnace for solid samples (Analytik Jena, Multi NC2100S; Rinaldi

et al., 2007) (DL = 37.9 µgC/filter). For the organic carbon (OC) analysis, the punch was acidified before analysis to remove inorganic carbon from TC and obtain OC. This process included positioning the punch in the instrument sample container, covered with 40 µL of $H_3PO_4$ (20% w/w) and left under oxygen flow, at room temperature, for ca. 5 minutes to allow the volatilization of carbonates as carbon dioxide ($CO_2$). The process was monitored online by the analyzer NDIR $CO_2$ detector: When the $CO_2$ level went back to baseline conditions, the vessel was placed into the furnace (950°C) for the OC analysis.

The water insoluble organic carbon (WIOC) is not measured directly but is derived from the difference between WSOC and OC. As mentioned, the measurements of OC and WSOC are made with two different instruments with that for OC having a much lower limit of detection (DL). Although, the quantification of WSOC was always possible, some samples had OC concentrations < DL. Assuming that OC=WSOC (and WIOC = 0) would be incorrect andwould result in significant error in the estimate of total OC. There could be an important amount of WIOC that we cannot quantify because of the lower sensitivity of the OC analysis. For this reason, we presented WIOC and OC data only for samples that have both quantifiable OC and WSOC.

25. **Specific comments: Abstract states 15 min time resolution or ACSM while it is 10 min in the text (line 244). Be consistent, and state what was the real time resolution.**

**The text has been corrected to 10 minute time resolution.**

26. **Provide reference for statement on lines 49-50 on the sea spray emission**

"Oceans cover approximately 70% of the Earth's surface and sea spray emissions are estimated to contribute from 3 to 30 Pgyr−1 to the global emitted particulate matter (Lewis and Schwartz, 2004), making them a major primary source in the atmosphere.

Lewis, E. R. and Schwartz, S. E.: Sea Salt Aerosol Production:Mechanisms, Methods, Measurements and Models – a Critical Review, Geophys. Monogr. Ser., vol. 152, AGU, Washington,D.C., 413 pp., 2004.

27. **Fig 1: provide units for Chl-a and POC**

This figure is now updated.

28. **Line 147: what was the time resolution for these measurements? Also, for other measurements of water composition like Synechococcus, PicoEukaryotes, NanoEukaryotes, Coccolithophore, Cryptophytes, etc.?**

The time resolution for all measurements is now included.

For Section 2.2.2: Chlorophyll-a/POC
Line 153: "Both the Chl-a and the POC were obtained with a time resolution of 1 minute."

For Section 2.2.3:Line 171: "These samples were collected at stations marked on the ship transect providing a total of 25 samples."

Section 2.3.1: Line 175: "Surface microlayer SML sampling was conducted twice a day from a zodiac using a 50 x 26 cm silicate glass plate sampler (Harvey 1966; Cunliffe and Wurl 2014) with an effective sampling surface area of 2600 cm$^2$ considering both sides."

29. Lines 225-226: provide more details on acidification before analysis or provide an appropriate reference to the method

Additional text has been included.

Line 244:" The punch was acidified before analysis to remove inorganic carbon from TC and obtain TOC. This process included positioning the punch in the instrument sample container, covered with 40 μL of H3PO4 (20% w/w) and left under oxygen flow, at room temperature, for ca. 5 minutes to allow the volatilization of carbonates as carbon dioxide (CO2). The process was monitored online by the analyzer NDIR CO2 detector: When the CO2 level went back to baseline conditions, the vessel was placed into the furnace (950°C) for the Total Organic Carbon analysis."

30. Lines 287-288: correct the sentence, it has 'showed' and 'is' in the same sentence

We have modified this phrase.

Line 321: "Mass concentration of aerosol chemical composition obtained from the submicron offline filter measurements are listed in Table 1. "

31. Line 312: why the real sea salt density was not used for the sea salt, which is 2.165 rather than 1.75. What other inorganics were there?

This is changed and results in a better comparison between the ACSM and the DMPS and in better agreement with the offline filters.

Updated text, Line 344." In order to determine how representative the ACSM PM1 measurements were of the total PM1 mass, the total ACSM PM1 mass concentration was converted into volume concentration (dividing organic mass concentrations by a density value of 1.2 g cm$^{-3}$, and the other remaining components SO$_4$, NH$_4$, Chl, and NaCl by 2.165 g cm$^{-3}$ (Seinfeld and Pandis, 2006))."

32. **Figure 4c: correct the labelling, it is either shifted or labels are given to minor m/z, difficult to read this figure.**

This is corrected

33. Line 371: provide the H/C value

The value was provided in the previous sentence, however, we included it again.

"The O/C ratio of this POA$_{NR}$ was 0.1 and the H/C was 1.64, which is typical for values of primary organic aerosol in the ambient atmosphere. The POA$_{NR}$ factor identified in this work, as well as the H/C ratios (1.64), was similar….."

34.  Lines 414-415: I'm confused with this statement, do you refer to satellite based or concurrently measured Chl-a or both?

**This phrase has been removed**

35.  **Line 415: I question the use of appropriate reference here as, to my knowledge, Rinaldi et al. 2013 did not discuss these issues. Provide the correct reference.**

**This phrase has been removed**
* * *
**Anonymous Referee #2**

**General comments**

The manuscript "Mediterranean nascent sea spray organic aerosol and relationships with seawater biogeochemistry" by Freney et al. presents aerosol composition data collected aboard a 5 week voyage in the Mediterranean Sea in the late-spring – early summer of 2017. The project continuously pumped bulk-seawater from 5m below the surface into a sea-spray generator which uses a plunging jet system to produce primary aerosol. Aerosols generated in the apparatus were then sampled continuously with a range of aerosol instrumentation – of which this paper focuses on the chemical composition measurements: the ToF-ACSM and filter-methods. Surface sea-water was also measured in parallel for phytoplankton cell count, chlorophyll-a, POC. Sporadic sampling of the sea surface microlayer was also performed manually, with DOC analysis undertaken on these samples.

Positive Matrix Factorisation was applied to the ACSM mass spectra which identified four significant factors making up the non-refractory organic aerosol (OA) include primary OA, oxidized OA, methanesulfonic acid type OA and mixed OA. Interesting correlations were found between these factors and phytoplankton cell abundance, as well as the POC concentrations. No relationship was observed with chlorophyll-a, but this was likely because of only minimal variance in this parameter throughout the voyage.

The manuscript is well written and provides an interesting dataset to help further understand the organic fraction of primary aerosol generated from bubble bursting. The authors thoroughly assess their data and generally speaking, identify well the limitations of the dataset, however a few conclusions in the discussion are a bit overstated. One of the main limitations of the paper is the model parameterisation section, which leaves much to be desired in terms of convincing the reader that these are robust and significant relationships.

Overall, I would recommend this paper be accepted after minor revisions.

Figures (Fig. 8) and discussion (Line 579 to 588) on the model parameterizations have been added to the manuscript (these can be seen above in response to comment no .19 of reviewer 1). This includes a discussion on the evaluation of the level of variance in the different organic classes explained by the parameterization, informing the reader about the robustness of these predictions.

**Specific scientific comments**

1.  Line 80 requires a reference

This phrase has been modified
"In marine environments, aerosol mass spectrometry has been used to better characterize the chemical properties of ambient marine aerosol particles, from the Atlantic (Ovadnevaite et al., 2011), to Antarctic coastal environments (Schmale et al., 2013, Giordano et al., 2017). "

2.      Line 176 – where did the waste-water from rinsing go? Was there any chance it could contaminate or disturb the microlayer before sampling?

The rinsing procedure was performed on the « Pourquoi Pas ?» with wastes evacuated at a different location than where sampling was performed (downstream of the sampling point). Sampling of the SML was made from a rubber boat that travelled to a distance from the « Pourquoi Pas ?», in order to avoid any contamination from the ship.

Added text Line 200: "The waste water was evacuated downstream of the sampling location to avoid any contamination."

Line 193 – what speed was the air flowing across the water?

The air speed across the seawater surface was 13 m-s-1. This is now specified in the text.
Section 2.4

3.      what are the models, manufacturers and setup parameters of the ACSM, DMPS, CPC and impactor?

We have updated the text to include the necessary information.
"The various aerosol instrumentation, including a time of flight aerosol chemical speciation monitor (ToF-ACSM, (Aerodyne)), a custom made differential mobility particle sizer (DMPS) coupled with a condensation particle counter (CPC, model 3010 (TSI)), and an impactor (Dekati,PM1) that collected submicron particulate matter for offline ion chromatography analysis, sampled from the headspace above the seawater in the tank. "

4.      What was the diameter and material of the sampling line? Did you characterize the sample losses? Tof-ACSM - did you use a standard or capture vaporizer?

This information is included in the text. Samples losses were not calculated. We did not report quantitative flux measurements during this study, we just report the evolution of the different species over time, as each species should have experienced similar losses.

Line 200: "A silica gel dryer was connected to the output of the chamber, which was subsequently connected to a flow dispatcher which had two outputs connected of equal length (<50 cm), one to the ToFACSM and the second to a SMPS-CPC. "

Line 262: "The ToF-ACSM contains a critical orifice, a PM1 aerodynamic lens to focus submicron particles into a narrow beam that flows into a differentially pumped vacuum chamber, a standard vapourizer heated (600$^{o}$C) to vaporize particles, an electron emitting tungsten filament (70eV) to ionize the vapor, a compact time-of-flight, mass analyzer (ETOF, TOFWERK AG, Thun, Switzerland) and a discrete dynode detector (Fröhlich et al., 2013)."

5.      Table 1 – what is the uncertainty on these? Is it 2 standard deviations, 95

The error is 1 SD and this is now included into the table caption.
"The cited uncertainty represents 1SD"

6.      Section 3.1 – could the discrepancy between the ACSM and filter measurement be more simply explained by the differences in upper diameter limits between the two measurements techniques (i.e. ACSM is 500 nm, presumably the filter is 1um)?

We believe that it is more likely to be a result of the refractory material because the DMPS (with an upper size cut of 500 nm) is in agreement with the filter measurements.  This is illustrated by Fig. S3 in the supplementary figures.

7.      I would like to see a graph of the size distributions, even in the supplementary information.

The average size distribution of sea spray generated during the experiment is shown in Fig. R5  and can be found in the supplementary material of Sellegri et al., 2020 (in press), because of copyright we are unable to provide this figure in this paper. The time series of the normalized size distribution is also shown as figure 1 of the Sellegri et al. (2020) paper. The time series shows that the shape of the size distribution does not change very much with time. Rather, it is the number concentration of the sea spray that changes with the seawater biogeochemical properties.

[Figure]

Fig. R5 Normalized size distributions in blue: arctic mesocosms, yellow: Mediterranean ship and orange: NZ coastal mesocosms. Bold line are for the full size decomposition and light lines for their mode decomposition. Sellegri et al., (2020)

8.      Line 336 – please include the reference for the "reference mass spectra"

The references for all the "reference mass spectra" are included in the following paragraph from lines 355. We did not make any changes to the text.

9.      Figure 4 – what are the 3 different pie graphs? Please label or include in the caption.

Three bars have been included into the figure to illustrate when the period over which the PIE chart data corresponds. The figure and the caption are updated.

10.      Figure 5 and its conclusions – the" error bars on this figure are very difficult to read, please consider offsetting them or altering the presentation here. Looking at this figure, complete with the error bars, I find it difficult to come to the conclusions the authors have about there being a diurnal cycle present at all. There is variability from hour to hour, but given the size of the standard deviation, none of this is significant enough to conclude any diurnal cycle.

Although the variation is very small, there is a systematic increase in the morning and afternoon concentrations for all species (except the MSA).

In the text, we indicate that there is only a slight variation and we now add that this variation is not significant; we therefore leave the statement that this small variation makes photochemical treatment unlikely.

Line 456: Although very weak and within the measurement error, $OOA_{NR}$ $MSA\text{-}OA_{NR}$, and $MOA_{NR}$ factors had a similar diurnal variation with increases during the early hours of the morning and again in the afternoon (Fig 5). $POA_{NR}$ species showed similar variations to the other species in the morning but a second increase was not observed in the afternoon. A lack of strong diurnal variation of $MSA\text{-}OA_{NR}$ and $OOA_{NR}$ indicates that the

secondary (oxygenated) nature of these compounds might be a result of biological processing rather than photochemical processes.

11.     While I commend the authors on their efforts to parameterise the production of organic aerosol from POC concentrations, I would like to see the actual underlying data supporting this from your study – i.e. the figures showing POC vs the fractions of OA, and how your models actually fit the data. I wouldn't put a lot of confidence in the equations without seeing the data, and even then, the Pearson coefficients are reasonably low so trusting and implementing this may be a stretch.

Figures (Fig. 8) and discussion (Line 579 to 588) on the model parameterizations have been added to the manuscript (these can be seen above in response to comment no .19 of reviewer 1). This includes a discussion on the evaluation of the level of variance in the different organic classes explained by the parameterization, informing the reader about the robustness of these predictions.

**Technical comments**

a.     Line 98 – name of vessel should be italicised. Does it need the "?"?OK, Yes the ? is part of the name
b.     Line 196 – "different" should be "various"OK
c.     Line 312 – remove space after "1.2 g cm-3"OK
d.     Line 313 – missing close parenthesis in sentence.. OK
**e.**     Figure 3 caption - please check units. **OK this is corrected.**
f.     Line 326 – remove full-stop before "(Fig S7)".OK
**g.**     Line 428 – remove semicolon in "Fig; 6"OK

---

## Author Response (AR2)

I thank the authors for the replies and the work they've put into it!

**Thank you again for taking the time to (re)examine our manuscript. Your input and opinions are appreciated. We have decided to remove the discussion on the refractory OM. As you pointed out, with the available information, there are too many uncertainties to make a solid conclusion. We have addressed each of your additional comments below.**

Nevertheless, the case presented for the refractory OM is still not convincing. The explanation provided does not exactly rule out the possibility of the overload by the sea salt. Indeed, concentrations up to 100 µg m-3 or higher can be measured by the instrument, but these refer to flash vaporising species. So, in normal conditions, where mass concentration is dominated by the non-refractory flash vaporising species, the instrument can measure such high concentrations, but it has never been shown that it is similarly capable of measuring such high concentrations of slow vaporising species. Quite on contrary. The fact that background (filter) OM did not show significant changes over the course of the campaign (Figure R1) does not exactly point to the lack of the overload (sea salt m/z's would be more applicable, like the background signal of NaCl at m/z 58 and Na at 23, to see the overload issue),

**Although there are uncertainties in interpreting the time series of the missing mass, and the hypothesis behind the attribution of this missing mass to organic matter, we believe that the ACSM measurements do not suffer from overloading of the instrument and provide additional information to back-up this discussion (also now included in the paper). First, the total concentrations measured by the SMPS simultaneously to the ACSM were on average 1400 # cc$^{-1}$ (75% were below 1600 # cc-1) which is very close to ambient concentrations that can be found in the Mediterranean atmosphere. Second, in the figure below we show the time series of the background signal at m58, in Figure a) the variation of the m58 is shown as a function of time. We observe that this signal varies with the total mass loading but does not increase progressively over time. In b) and c) two zero particle periods are shown from the start and the end of the campaign. The filter periods show that the zero particle periods did not become progressively higher from the start to the end of the campaign.**

[Figure]

**Figure 1 Time series of the filter m/z 58 over the a) whole time period, b) over a filter period at th start of the campaign and c) at the end of the campaign.**

**We have added the following text to the discussion in Section 3.1 (lines 380-387 in the current version):**

« We ruled out the possibility that the high fraction of inorganics in the SSA led to an accumulation of refractory or semi-refractory material on the vaporizer and a corresponding decrease in the ability to measure non refractory material by examining the background (filter) signals. Figure S11 shows the background signal at m/z 58 (NaCl$^+$) as a function of time; this signal varies as a function of total mass loading but does not progressively increase over time. Similarly, particle free sampling periods at the start and the end of the field campaign show that m/z 58 background levels dropped to comparably low values. Therefore, we conclude that overloading of the ACSM by refractory, or semi-refractory slowly vaporizing, material did not occur and the missing mass is due to increased refractory content. »

but, actually, it shows that there was no significant amount of refractory OM. E.g. if refractory OM contribution is so much higher in the second part of the campaign would it not manifest in the higher background OM concentrations? But now, the OM behaves exactly as SO4 for the part when refractory OM is not anticipated and the part where it is?

**We assume that by refractory OM you mean slowly vaporizing OM. We agree that the refractory mass cannot be unambiguously attributed to organic matter and have removed the discussion on the refractory OM.**

Moreover, looking at figures 3, S3 and S13, I struggle to see the coherent picture. Why S3 has fewer points for filter than S13? E.g. S13 has filter OM data since 16th May, while in S3 filter points start after 19th?

**These three figures are updated, to include only data points from the 17$^{th}$ onwards. Prior to this period our Airbeam values were unstable**

Big emphasis is attributed to the discrepancies between ACSM and Filter/DMPS at the end of the campaign, however, there are similar if not larger discrepancies between filter and ACSM OM at the beginning of the campaign (16th-21st May) and these are not reflected in the volume comparisons. The filter and ACSM volumes seem to agree for the period where Filter-ACSM OM disagrees by a lot (16th-21st May). While looking at Figures 3 and S3, it seems that ACSM volume is larger than DMPS volume for the period where ACSM OM is much lower than the filter OM (19th-20th May)?

**We are aware of this discrepancy, where the AMS volume comparisons are similar at the start but that the organic aerosols do not agree with the filters. However, the OM concentrations are only a small part of the total mass so we do not think that this would necessarily impact the total volume concentrations.**

Also, the 'oversampling' (~19th May and ~25th-29th May) by ACSM is not discussed anywhere, which is now on the order of the 'under-sampling' later in the campaign (after the 1st June). The latter is attributed to the 'refractory' OM, so how do authors explain the former (oversampling)? Is that just uncertainty (both) due to density assumptions?

**An incorrect calibration value was introduced when revising these two figures in the last version of the paper. This has been corrected, and the difference between ACSM and SMPS volumes during the periods of 19$^{th}$ May and 25-29$^{th}$ May are within the uncertainties of the measurements, (including, as the reviewer suggested, the uncertainty on density).**

All these would make me very cautious in attributing the discrepancies between ACSM and DMPS to the refractory OM. It is a bit like cherry picking – authors attribute under sampling to refractory but ignore oversampling. Also, ignore the fact that high OM discrepancies between filter and ACSM still lead to good agreements in volumes at the first part of the campaign…To me, the explanation of

different sensitivity to sea salt for distinct water composition and different sea salt ions, would be much more feasible explanation than the refractory OM.

The volume/volume discrepancies trends do not coincide with OM composition (neither from filters as in WSOC/WIOC trends nor PMF as in trends for different factor contributions), which really makes the refractory OM explanation quite unlikely. Also, the ssSO4 behavior mimics that of 'refractory' OM, which points again to the sea salt rather than OM. More so that OM is more oxidised for the later part of the campaign.

**The reviewer has made a number of valid points. Although we have kept the discussion comparing the DMPS with the ACSM and state that there is missing mass, we have removed all discussion of this missing mass being attributed to refractory organic fraction. We have replaced the observed relationship between the refractory organic matter and POC with the observed relationship between the offline filter OM and the POC.**

**The following text is added (lines 592-596 in the current version:**

« We initially examined the relationship with offline filter organic measurements of water insoluble and water soluble organic matter fractions (FWIOM and FWSOM, respectively) with POC (Eq. 2 and 3). These showed positive correlations of both WIOM and WSOM with POC measurements.

FWSOM =0.002 [POC] -0.0393              r = 0.55, n = 19, p < 0.01                          Eq.2

FWIOM = 0.002 [POC] + 0.031            r = 0.54, n = 12, p <0.05                          Eq. 3 »

On a minor note:
Figure S3 b) is difficult to read as points are not properly represented in the legend. Which ones are the ACSM and which ones are the Filter?
**The figure is updated.**

Also, density of 2.165 is applicable to sea salt, but not other inorganic species. E.g. NH4 (line 350 of the revised manuscript). Presence of NH4 would mean the presence of nss-SO4 in addition to ssSO4? Were these significant or on the level of noise? Do not include them in the calculations if the latter.

**The NH4 and NO3 are close to the zero during this campaign, so we don't think that this would matter. However, we recalculated the volume concentration without contributions from these species. The text and figures are updated.**

Reply No 9: 30% noise is very high; this is not typical for PMF. Does that 'noise' have meaningful time trend or mass spectrum, could it be some unresolved OM factor?

**No, we believe that this high noise was a result of low mass concentrations and low temporal evolution of the mass spectra. It may be some unresolved OM but the contribution remained constant throughout the sampling.**

Reply 13: 'In addition, during sampling the ToF-ACSM was run with a 2 min filter/8 minute sample. When sampling with long times between filters, any drift in sensitivity can result in a difference signal that is an artefact.' – I would not expect any significant drift in sensitivity for 8 minute period? Where can it come from? Change in SEM, filament? They would not occur on the 8 minute time scale. Provide reference or do not refer to it in the manuscript.

**Unfortunately there is no reference for this observation. This was concluded from direct observation and discussion with other co-authors and Aerodyne. The sensitivity of the detector in the ToF-ACSM is quite temperature dependent and we do see fluctuations of several percent over a few minutes in response to room temperature changes. Unfortunately, this can cause an artefact at m/z's that have high backgrounds due to air or water.**

Some typos and missing spaces should be corrected, especially for the newly added text.

**We have read over the manuscript and have corrected these errors.**

---

## Author Response (AR3)

**Mediterranean nascent sea spray organic aerosol and relationships with seawater biogeochemistry, Freney et al., ACP_2020_406**

Dear Editor/ reviewer,

This was an oversight on our behalf and we missed this sentence in the abstract. This sentence has been removed.

Thank you again for taking the time to review the manuscript.

Best wishes

Comments to the Author:
Thank you very much for the careful revision of your manuscript according to the reviewer's comments. However, the abstract of the manuscript needs also to be corrected accordingly, as noticed by the reviewer (see comments). After this last correction the manuscript can be published in ACP.
kind regards,

I appreciate the revisions that were made by authors and responses are sufficiently ok to proceed with a publication, subject to one final revision.

The abstract (lines 44-46) states that 'Filter-based analysis of the submicron SSA showed that the non-refractory organic mass represented on average only 40% of the total organic mass, which represents approximately 22% of the total sea spray mass.' – to my understanding, the 'refractory OM' was derived from filter-ACSM comparison and not the filter-based analysis as stated above. I will not go again into the argument that the discrepancy between filter OM and ACSM OM does not necessarily mean refractory OM (especially that the largest discrepancies, i.e. 'refractory OM', occur for the periods where the agreement between total volumes of filter/DMPS and ACSM is good).
The data show that ACSM measured ~40% of the filter OM, but this should not be attributed to 'refractory OM' in the abstract.